# RF-Policy: Rectified Flows are Computation-Adaptive Decision Makers

## Abstract

Diffusion-based imitation learning improves Behavioral Cloning (BC) on multi-modal decision-making but comes at the cost of significantly slower inference due to the recursion in the diffusion process. However, in real-world scenarios, states that require multi-modal decision-making are rare, and the huge consumption of diffusion models is not necessary for most cases. It inspires us to design efficient policy generators that can wisely allocate computation for different contexts. To address this challenge, we propose RF-Policy (Rectified Flow-Policy), an imitation learning algorithm based on Rectified Flow, a recent advancement in flow-based generative modeling (Liu et al., 2022). RF-Policy adopts probability flow ordinary differential equations (ODEs) for diverse policy generation, with the learning principle of following straight trajectories as much as possible. We uncover and leverage a surprisingly intriguing advantage of these flow-based models over previous diffusion models: their training objective indicates the uncertainty of a certain state, and when the state is uni-modal, they automatically reduce to one-step generators since the probability flows admit straight lines. Therefore, RF-Policy is naturally an adaptive decision maker, offering rapid inference without sacrificing diversity. Our comprehensive empirical evaluation shows that RF-Policy, to the best of our knowledge, is the first algorithm to achieve high performance across all dimensions, including success rate, behavioral diversity, and inference speed.

## 1 Introduction

Imitation Learning (IL) is prevalent in robot learning for addressing continuous control challenges. Unlike Reinforcement Learning (RL), which requires the manual specification of a reward function, IL is particularly well-suited for learning complex, "non-declarative" motions. The go-to method for IL is Behavioral Cloning (BC), where an agent performs supervised learning to acquire a policy $\pi$ mapping states to actions.

While BC is straightforward to implement and quick to train, it is limited in terms of behavioral diversity (Mandlekar et al., 2021; Shafiullah et al., 2022; Chi et al., 2023; Florence et al., 2022). Specifically, since BC learns a deterministic mapping, it can struggle with one-to-many relationships, a common scenario where an agent can perform multiple actions to solve a task at a given state. Recent advancements in diffusion models, originally successful in generative image modeling, have been adapted for policy learning (Chi et al., 2023; Ajay et al., 2022). These models offer greater behavioral diversity but are less prevalent in real-world robotic learning tasks due to computational inefficiencies. For instance, training diffusion-policy necessitates several weeks, as indicated in Chi et al. (2023), in contrast to BC which only requires a matter of hours Mandlekar et al. (2021). Moreover, during execution, diffusion models need to simulate a stochastic differential equation, which makes it slow in execution as well.

In practical applications of robotics, we find that demonstration data frequently exhibit a unique characteristic: in the majority of the states, the actions are uni-modal and close to deterministic given the state for which a simple and fast BC algorithm is sufficient, while a small portion of critical states requires diverse and multi-modal actions given a single state, whose behavior need to be captured by more expensive diffusion models. Consequently, an ideal imitation learning algorithm should

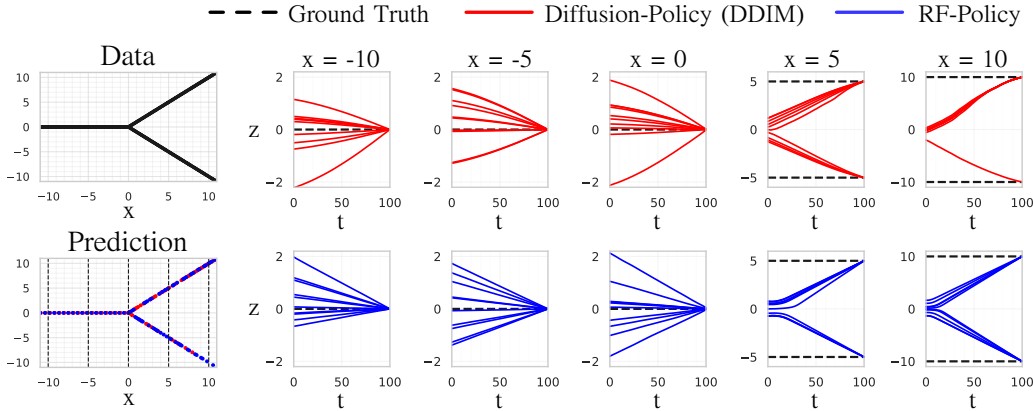

Figure 1: Illustrating the computation adaptivity of Rectified Flow. We use DDIM and RF-POLICY to predict $y$ (action) given $x$ (state), with deterministic $y = 0$ when $x \leq 0$, and bimodal $y = \pm x$ when $x > 0$. Both DDIM and RF-POLICY fit the demonstration data well. However, the simulated ODE trajectory learned by Diffusion-Policy with DDIM (red) is not straight no matter what $x$ is. By contrast, the simulated ODE trajectory learned by RF-POLICY (blue) is a straight line when the prediction is deterministic ($x \leq 0$), which means the generation can be exactly done by one-step Euler approximation.

automatically identify and rapidly learn from the uni-modal segments of the data, while spend more computation on multi-modal part to capture the behavioral diversity therein.

To address the above requirements, we introduce a novel ODE-based imitation learning algorithm, denoted as RF-POLICY, which brings the gap between BC and diffusion-based methods. Like BC, RF-POLICY employs a straightforward supervised learning objective, ensuring the simplicity of implementation and fast training. Like diffusion models, it learns a rich multi-step ODE (or flow) model to fully capture the action diversity in states with multi-modal behaviors. What makes RF-POLICY unique is it is automatically adapted to the diversity of the actions, by learning ODEs with straighter trajectories, hence faster to simulate, on states with more deterministic actions.

Along with RF-POLICY, we also introduce a novel set of metrics for a more comprehensive evaluation of an imitation learning algorithm, namely, the precision (success rate), the recall (behavioral diversity), the training efficiency, and the execution efficiency. Through a comprehensive evaluation across a toy domain, as well as robot manipulation learning problems, we demonstrate that RF-POLICY is, to our knowledge, the first IL method that achieves uniformly good performance across all of the above criteria.

To summarize, our contributions are:

- We propose a novel offline IL method, RF-POLICY, that is simple to implement, fast to learn and execute, and automatically captures demonstration diversity only when necessary.

- We provide theoretical evidence showing why RF-POLICY reduces to a one-step model (just like BC) at states with uni-modal behavior.

- We introduce new metrics for comprehensive evaluation of an IL algorithm, and conduct a thorough evaluation across multiple robotics problems to showcase the benefit of RF-POLICY. RF-POLICY performs consistently and uniformly better than previous methods.

## 2 BACKGROUND

**Imitation Learning** Imitation Learning (IL) is a paradigm in machine learning where the agent learns optimal policies by mimicking the behavior of an expert, without explicit reward feedback (Schaal, 1999). The fundamental idea is to approximate the expert's decision-making process, represented by the mapping of observed states to actions. The training dataset comprises pairs of states and the corresponding expert actions $\{(x^{(i)}, y^{(i)})\} \sim p^*$, where $x$ represents the state, $y$ denotes the expert action, and $p^*$ is the unknown true distribution of state-action pairs.

Behavioral Cloning (BC) (Torabi et al., 2018; Mandlekar et al., 2021) is a widely-used method in IL, which employs supervised learning to learn a deterministic policy $\pi$ that maps states to actions.

However, it primarily estimates the conditional expectation of the actions and struggles with scenarios where the state-action relationship is stochastic, non-Gaussian, or multi-modal. The limitation of BC in handling such scenarios motivates the exploration of more expressive models, such as diffusion models (Chi et al., 2023; Janner et al., 2022; Ajay et al., 2022).

**(Conditioned) Rectified Flow**   Given a set of $N$ data points $\{(x^{(i)}, y^{(i)})\}_{i=1}^{N} \sim p^*$ drawn from an unknown distribution $p^*$. The goal is to estimate the conditional distribution $p^*(\ y \mid x\ )$ of the output $y$ given $x$, in the form of an $x$-conditioned generative model. This is highly challenging task because given each $x$, $p^*(y|x)$ can be non-deterministic, non-Gaussian, and multi-modal. In contrast, the standard regression techniques, for example, only estimates the conditional expectation $f(x) = \mathbb{E}_{p^*}[y|x]$, but fails to capture the whole distribution and sample from it.

We approach this problem with the (conditional) rectified flow framework of Liu et al. (2023b); Liu (2022), in which we learn a conditional ODE (or flow) model

$$\frac{\mathrm{d}}{\mathrm{d}t} Z_t = v(Z_t, t \mid x), \tag{1}$$

where the conditional velocity field $v$ is parameterized as a neural network. We should learn $v$ from the data such that when we start from an initial $Z_0 \sim \pi_0$ at time $t = 0$, the flow model outputs a $Z_1$ at time $t = 1$ that obeys $Z_1 \sim p^*(\cdot|x)$. $\pi_0$ is an elementary noise distribution, usually the standard Gaussian distribution.

Ideally, if we observe a pair of $(x, y)$, an ODE that achieves $z_1 = y$ given $x$ would be simply $\frac{\mathrm{d}}{\mathrm{d}t} Z_t = (y - Z_0)$, which travels along the straight line from $Z_0$ to $y$ with uniform speed, yielding a closed form $Z_t = ty + (1 - t)Z_0$. In the inference time, the oracle ODE is impractical since $y$ is unknown, but we can learn $v$ to approximate the oracle ODEs in the training set as much as possible,

$$L(v^*; x) = \min_v \mathbb{E}_{(x,y)\sim p^*, Z_0 \sim \pi_0} \left[ \|y - Z_0 - v(Z_t, t \mid x)\|_2^2 \right]. \tag{2}$$

The special aspect of the objective is that the oracle ODEs are constructed by connecting straight lines, which gives it unique advantage as we will discuss in Section 3.

**Connection with Other Diffusion / Flow Models**   Other classic probability flow ODEs, like DDIM (Song et al., a), can be viewed as non-linear rectified flows. However, their oracle ODEs do not necessarily travel in straight trajectories with uniform speed. Formally, they admit a general form of

$$Z_t = \alpha_t y + \beta_t Z_0, \quad \frac{\mathrm{d}}{\mathrm{d}t} Z_t = \dot{\alpha}_t y + \dot{\beta}_t Z_0 \tag{3}$$

where $\alpha_t$ and $\beta_t$ are two time-differentiable sequences satisfying $\beta_0 = \alpha_1 = 1$ and $\beta_1 = \alpha_0 = 0$. These oracle ODEs can be curved depending on the different choices of $\alpha$ and $\beta$. Fitting them yields the following objective,

$$\min_v \mathbb{E}_{(x,y)\sim p^*, Z_0 \sim \pi_0} \left[ \left\| \dot{\alpha}_t y - \dot{\beta}_t Z_0 - v(Z_t, t \mid x) \right\|_2^2 \right]. \tag{4}$$

Previous probability flow ODEs, e.g., DDIM (Song et al., a), VP-ODE, sub VP-ODE (Song et al., b), can be recovered with specific choices of $\alpha_t, \beta_t$. Please refer to Liu et al. (2023b) for details.

## 3   RF-POLICY

### 3.1   EULER APPROXIMATION AND THE NECESSITY OF STRAIGHTNESS

Rectified Flow is naturally effective in generating results when managing deterministic relationships. To clarify terminology, we will be referring to "variance" as the measure of variability or uncertainty in the predictions made by the model. In Rectified Flow, we use the term 'straightness' to describe whether the trajectory that the ODE model $\frac{\mathrm{d}}{\mathrm{d}t} Z_t = v(Z_t, t \mid x)$ traces from the initial state $Z_0$ to the final state $Z_1$ in the ODE is following *straight-line* paths. In this case, a *single* step of Euler update allows us to calculate the output $Z_1$ from $Z_0$:

$$Z_1 = Z_0 + v(Z_0, 0 \mid x).$$

In comparison, ODEs with curved trajectories would require to run Euler or other discretization algorithms for multiple steps and with a small step size, causing a slow inference time.

In the following, we present a main theorem that shows that, when minimize the Rectified Flow objective exactly, the learned ODE is automatically straight on the states with deterministic actions, hence yielding fast inference time. We write $\text{var}(y \mid x) = \mathbb{E}[\|y - \mathbb{E}[y \mid x]\|_2^2 \mid x]$ as the conditional variance.

**Theorem 1.** *If* $\text{var}(y \mid x) = 0$, *then* $L(v^*; x) = 0$, *and the trajectories of ODE by* $v^*(\cdot; x)$ *are straight. In other words, whenever the conditional prediction is deterministic, Rectified Flow will render exact straight-line trajectories.*

*Proof.* In Rectified Flow, we optimize

$$\min_v L(v) = \int_t \mathbb{E}[\|y - Z_0 - v(Z_t, t; x)\|^2] dt, \quad \text{where } Z_t = ty + (1-t)Z_0, Z_0 \sim \pi_0, (x, y) \sim p^*. \tag{5}$$

The optimal solution of (5) has been shown in Liu (2022) to be

$$v^*(z, t \mid x) = \mathbb{E}[y - Z_0 \mid Z_t = z, x]. \tag{6}$$

As a result,

$$L(v^*, x) = \mathbb{E}_{(x,y)\sim p^*, Z_0 \sim \pi_0}[\|y - Z_0 - v^*(Z_t, t \mid x)\|_2^2] = \int \text{var}(y - Z_0 \mid Z_t, x) dt. \tag{7}$$

Since $Z_t = ty + (1-t)Z_0$, we have

$$y - Z_0 = y - \frac{Z_t - ty}{1-t} = -\frac{y - Z_t}{1-t}.$$

But then,

$$\text{var}(y - Z_0 \mid Z_t, x) = \text{var}\left(\frac{y - Z_t}{1-t} \mid Z_t, x\right) = \frac{1}{(1-t)^2} \text{var}(y \mid x) = 0.$$

Plugging the above into (7), we have $L(v^*, x) = 0$. $\qquad\square$

The motivation behind ensuring straight trajectories in Rectified Flow (RF) stems from the intrinsic characteristics of ODEs. ODEs are computationally demanding, especially when dealing with intricate trajectories, thus necessitating the pursuit of more straightforward paths for efficient and rapid generation. When a trajectory is straight, it can be simulated with a smaller number of iterations, significantly reducing the computational load and expediting the model's output. This is particularly beneficial when dealing with deterministic relationships, as theorem 1 suggests, where the variance $var(y|x)$ is zero, leading to straight trajectories.

In Figure 1, we use a straightforward toy example to illustrate this property. We present a 1D example depicting the trajectories learned by both DDIM and our method (Rectified Flow) for different mappings of $y$. This illustration clearly demonstrates that, unlike the DDIM, Rectified Flow ensures straight trajectories when the prediction is deterministic, which validates its capability for efficient generation through one-step Euler approximation.

**Theorem 2.** *Let* $\pi_1$ *be the target probability distribution at time* $t = 1$, *that is,* $\pi_1 = \text{Law}(z_1)$. *Let* $N = 1/\epsilon_{\min}$ *represent the maximum number of steps allowed in the worst-case scenario. Then, the Wasserstein-2 distance between the distribution of* $Z_1$ *(or* $y$*) and* $\pi_1$ *is bounded as follows:*

$$W_2(\text{Law}(Z_1), \pi_1) \leq N\eta,$$

*where* $W_2$ *denotes the Wasserstein-2 distance.*

*Proof.* Consider the process of advancing the Rectified Flow with a step size $\epsilon$ from $z_t$, resulting in $\tilde{z}_{t+\epsilon} = z_t + \epsilon v(z_t, t)$. The expected squared $L_2$ norm of the difference between $\tilde{z}_{t+\epsilon}$ and the actual next state $Z_{t+\epsilon}$ is:

$$\text{err} = \mathbb{E}[\|\tilde{z}_{t+\epsilon} - Z_{t+\epsilon}\|_2^2 \mid Z_t = z_t].$$

Following Theorem 1, define the variance function $\sigma^2(z) = \mathbb{E}[\|y - Z_0 - v(Z_t, t)\|_2^2 \mid Z_t = z] = \text{var}(y - Z_0 \mid Z_t = z)$, which captures the variability of the discrepancy between the RF-predicted and actual values of $y$ given $Z_t = z$.

---

**Algorithm 1** RF-POLICY Two-Stage Training

---

**Input**: Demonstration data $\mathcal{D} = \{(s_i, a_i)\}_{i=1}^N$, learning rate $\alpha$, training iterations $K$, initial parameters $\theta_0, \phi_0$.
**Stage 1: Optimize Rectified Flow**
**for** k = 1: K **do**
  Sample data batch $\{(s_i, a_i)\}_{i=1}^B \sim \mathcal{D}$, and $\forall i$, time $t_i \sim U(0, 1)$ and noise $Z_{i,0} \sim \mathcal{N}(0, I)$.
  Compute

$$L_1(\theta) \approx \frac{1}{B} \sum_{i=1}^B \left( \frac{1}{2} \|a_i - Z_{i,0} - v_\theta(Z_{i,t_i}, t_i \,; s_i)\|^2 \right),$$

  where $\forall i$, $Z_{i,t} = t a_i + (1-t) Z_{i,0}$, Then update

$$\theta_k \leftarrow \theta_{k-1} - \alpha \nabla_\theta L_1(\theta_{k-1}).$$

**end for**
**Stage 2: Optimize the Variance Estimation Network**
**for** k = 1: K **do**
  Sample data batch $\{(s_i, a_i)\}_{i=1}^B \sim \mathcal{D}$, and $\forall i$, time $t_i \sim U(0, 1)$ and noise $Z_{i,0} \sim \mathcal{N}(0, I)$.
  Compute

$$L_2(\phi) \approx \frac{1}{B} \sum_{i=1}^B \left( \frac{1}{2\sigma_\phi^2(Z_{i,t_i}; s_i)} \|a_i - Z_{i,0} - v_{\theta_K}(Z_{i,t_i}, t_i \,; s_i)\|^2 + \log \sigma_\phi(Z_{i,t_i}; s_i) \right),$$

  where we treat the purple part as a constant. Then update

$$\phi_k \leftarrow \phi_{k-1} - \alpha \nabla_\phi L_2(\phi_{k-1}).$$

**end for**

---

**Algorithm 2** RF-POLICY Execution.

---

**Input:** current state $s$, minumum step size $\epsilon_{\min}$, error threshold $\eta$.
Initialize noise $z \sim \mathcal{N}(0, I)$, $t = 0$.
**while** $t < 1$ **do**
  Compute step size $\epsilon_t \leftarrow \min\left(\frac{\eta}{\sigma_\phi(z_t; s)}, 1 - t\right)$, and $\epsilon_t \leftarrow \max(\epsilon_{\min}, \epsilon_t)$.
  Update $z_{t+\epsilon_t} \leftarrow z_t + \epsilon_t v_t(z_t, t \,; s)$
  Update $t \leftarrow t + \epsilon_t$.
**end while**
Execute action $a \leftarrow z_1$.

---

Then, the error can be expressed as:

$$\text{err} = \mathbb{E}[\|\epsilon(y - Z_0) - \epsilon v(X_t, t)\|_2^2 \mid Z_t = z_t] = \epsilon^2 \sigma^2(z_t).$$

By choosing $\eta$ such that $\epsilon \leq \eta/\sigma(z_t)$, we ensure that:

$$\text{err} \leq \eta^2.$$

Considering that at each step, the error contribution is at most $\eta^2$, and with a maximum of $N$ steps, the accumulated discrepancy measured by the Wasserstein-2 distance is bounded by $N\eta$. This concludes the proof.

*Note:* Although the proof is provided based on the unconditional RF framework, it remains valid for the conditional case once we incorporate the condition $s$ without loss of generality. □

### 3.2 Learning Variance for Adaptive Decision Making

In practical applications such as robotics, having an algorithm that can efficiently handle both uni-modal (given an $x$, the prediction $y$ is deterministic) and multi-modal (the same condition $x$ can lead to multiple possible $y$) data is crucial. The key observation (or assumption) we make in this work is:

**Assumption 1.** *In typical imitation learning demonstration datasets for robotics tasks, the majority part of the data is uni-modal, and only a small portion of the data is muli-modal.*

The consequence of the Assumption 1 is that while diffusion-based policies could in principle model both uni-modal and multi-modal data, as the majority of the data is uni-modal (this is why BC works well in practice), the expense of training and executing large diffusion-models might overweigh the benefit it brings in robot learning.

Theorem 1 implies that when the relationship between input and output is deterministic, Rectified Flow can form a straight trajectory, simplifying the learning process and enhancing the generation speed. The property of variance equaling straightness in Rectified Flow allows the model to be efficient and rapidly learn uni-modal segments while preserving and learning the behavioral diversity in multi-modal parts, making it suitable for practical applications in robotics where such characteristics are commonly exhibited in demonstration data.

In robotics tasks, where actions are generated based on given states, the estimated variance can be indicative of the "straightness" of the rectified flow generation process: If the variance is low for a state, it implies that the system's output is fairly deterministic and follows a straight trajectory. Fewer steps in the ODE simulation might be sufficient to accurately model the robot's behavior, and hence improve the model's efficiency in inference. In order to identify these states, we propose to use the following objective:

$$\min_{v,\sigma} \int_t \mathbb{E}_{(s,a)\sim\mathcal{D},Z_0\sim\mathcal{N}(0,I)} \left[ \frac{1}{2\sigma^2(s)} \|a - Z_0 - v(Z_t,t \mid s)\|^2 + \log \sigma(s) \right] dt. \tag{8}$$

The optimization of this loss function results in an optimal estimate for the variance $\sigma^2(x)$, which is dependent on the state $x$. We found that in practice, learning variance estimation network separately from learning RF can better ensure stability. See Algo. 1.

Once the variance is estimated, it is used to adaptively determine the step size $\delta t$ for

numerically simulating the ODE. The step size determines how the state $x_t$ is updated at each step in the simulation. Specifically, we follows Algo. 2 during inference. Note that we can derive the threshold $\eta$ from the training data. For instance, we are able to regulate the proportion of states requiring a full ODE simulation, with the remainder being addressed through a single-step ODE approach.

## 4 Related Work

**Diffusion/Flow-based Generative Models and Adaptive Inference** Diffusion models (Sohl-Dickstein et al., 2015; Ho et al., 2020; Song et al., b; Song & Ermon, 2019) succeed in various applications, e.g., image/video generation (Ho et al.; Zhang et al., 2023; Wu et al.; Saharia et al., 2022), audio generation (Kong et al.), point cloud generation (Luo & Hu, 2021a;b; Liu et al., 2023b; Wu et al., 2023), etc.. However, numerical simulation of the diffusion processes typically involve hundreds of steps, resulting in noticeable slowness. Post-hoc samplers have been proposed to solve this issue (Karras et al.; Liu et al., 2021; Lu et al.; 2022; Song et al., a; Bao et al., 2021) by transforming the diffusion process into marginal-preserving probability flow ODEs, yet they still use the same number of inference steps for different states. Although adaptive ODE solvers, such as adaptive step-size Runge-Kutta (Press & Teukolsky, 1992), exist, they cannot significantly reduce the number of inference steps.

Recently, new methods (Liu et al., 2022; Liu, 2022; Lipman et al., 2022; Albergo et al., 2023; Albergo & Vanden-Eijnden, 2022; Heitz et al., 2023) have emerged, directly learning probability flow ODEs by constructing linear interpolations between two distributions emerge recently. Empirically, these methods exhibit more efficient inference due to their preference of straight trajectories. In our

work, we unveil a previously overlooked feature of these flow-based generative models: they act as one-step generators for deterministic target distributions, and their variance indicates the straightness of the probability flows for a certain state. Leveraging this feature, we design RF-POLICY that can automatically decide the number of inference step for different states.

**Diffusion Models for Decision Making**   For decision making, diffusion models obtain success as in other applications areas (Kapelyukh et al., 2023; Yang et al., 2023; Pearce et al., 2022). In a pioneering work, Janner et al. (2022) proposed "Diffuser", a planning algorithm with diffusion models that outperforms counterparts like Behavioral Cloning. Later, this framework is extended to tasks like offline reinforcement learning (Wang et al., 2022), visuomotor policy learning on physical robots (Chi et al., 2023). Later, Ajay et al. (2022) improves the framework by modeling policies as conditional diffusion models. Despite the success of adopting generative diffusion models as decision makers in previous works, they also bring redundant computation, limiting their application in real-time, low-latency decision making for autonomous robots. RF-POLICY propose to leverage rectified flow instead of diffusion models, facilitating adaptive decision making for different states while significantly reducing computational requirements.

## 5   EXPERIMENTS

We conducted a series of experiments to comprehensively compare RF-POLICY with Behavioral Cloning (BC) and existing diffusion-based imitation learning methods, considering four essential dimensions: precision, which represents task performance; recall, indicating behavior diversity; and training and inference efficiency. Our experimentation began with a 2D Navigation problem, offering enhanced visualization of the behavior of the learned policies. This was followed by an exploration into robotics, where we conducted imitation learning problems using the LIBERO benchmark.

**Implementation Details.** In our experiments, the architecture aligned with the CNN-based structure detailed in Diffusion Policy Chi et al. (2023). This alignment resulted in an action horizon set at 8 steps and the adoption of position control in the action space. In our studies, **BC** was implemented as a benchmark, applying behavior cloning in its most straightforward form and utilizing a Mean Squared Error loss function between the predicted and ground truth actions. The implementations for DDPM and DDIM remained consistent with those outlined in Chi et al. (2023). Across all experiments, consistency was maintained regarding architecture, input, and output, with all methods adhering to a similar experimental pipeline. In RF-POLICY, the network employed for variance estimation network is constituted of a 4-layer MLP. This network ingests inputs are the features extracted by the observation encoder, which is frozen and shared with the policy network. In practice, the variance estimation network takes the same input as rectified flow model, so we simply takes the intermediam feature from the rectified flow model, and then finetune a several layuer

### 5.1   EXPERIMENT SETTING AND EVALUATION METRICS

While task success rate (or return) is the most commonly utilized metric in the literature, we claim that it does not entirely encapsulate the performance of a learned policy. This inadequacy stems from the fact that task success rate predominantly measures the capability of a policy to achieve a task, potentially overlooking the behavioral diversity, a crucial aspect given the prevalence of multi-modal decision-making scenarios in practical applications. Hence, we propose to measure the models with four essential dimensions. For each experiment, evaluations are conducted over 50 episodes.

**Precision (Success Rate)**   . To assess precision, we randomly sample start points for the agent and evaluate the rate of successfully completed episodes.

**Recall (Diversity).**   The diversity between demonstration trajectories and generated rollouts is quantified using a specifically devised diversity score. The diversity score is computed as the proportion of observations in the demonstration trajectories that are closer to any observation in the rollout trajectories than to any other observation (not from the same trajectory) in the demonstration trajectories, mathematically represented as score $= \frac{1}{n} \sum_{i=1}^{n} \mathbb{1}(d(\text{demo}_i, \text{rollout}) <$ $\max(d(\text{demo}_i, \text{demo}_{-i})))$, where $d$ denotes the $\ell_2$ distance, $\text{demo}_i$ is the $i^{th}$ observation in the demonstration, rollout represents the rollout observations, $\text{demo}_{-i}$ denotes all demonstration observations excluding $\text{demo}_i$, $n$ is the total number of observations in the demonstration, and $\mathbb{1}(.)$ is

the indicator function. This score provides a measure of the diversity of the generated rollouts with respect to the demonstrations, shedding light on the model's ability to generate diverse trajectories. During diversity evaluation, the agent's start point is fixed, and the diversity of the generated 50 trajectories is measured.

**Training and Execution Efficiency**   In order to provide a holistic perspective of the model's practicality, training efficiency is measured, focusing on the computational resources and time consumed during the training phase. Similarly, execution efficiency is assessed, concentrating on the model's ability to perform tasks rapidly and efficiently during real-time implementation, thereby indicating its suitability for real-world applications. Efficiency metrics include the Number of Function Evaluations (NFE), which tracks the count of network inferences per action generation, and the Learning Efficiency Index (LEI), which quantifies the success rate over the course of training through the area under the curve.

## 5.2 NAVIGATING A 2D MAZE

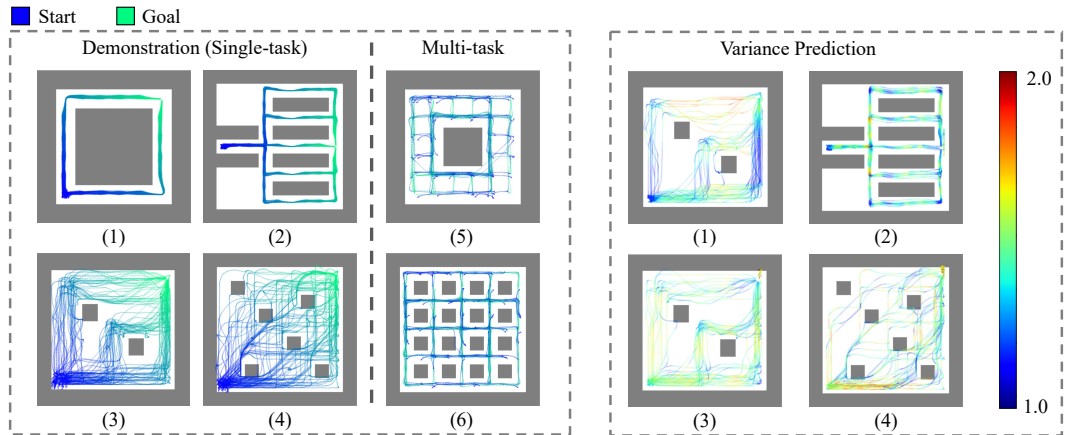

Figure 2: **Left:** Trajectories of 100 demonstrations for each maze. **Right:** Variance estimation by RF-POLICY, which indicates the diversity of generated actions or trajectories in a given state. We normalize the variance to $[0, 1]$ in each maze.

In this subsection, we describe experiments conducted in a simulated maze environment designed for agents to perform explorative tasks. We set up two different experimental scenarios:

In the first setting, the agent starts at a predetermined location, subject to minor random variations, and aims to navigate towards a fixed target. The objective here is for the agent to develop a policy that facilitates successful navigation. In the second scenario, the agent begins from a randomly assigned location and moves towards a random target. In this case, we devise a conditional policy, wherein actions are generated based on the given target. The goal-conditioned maze is inherently multi-modal, as every trajectory in the demonstration has different starts and ends.

The construction of the maze is based on D4RL Maze2D Fu et al. (2020), simulated by MuJoCo. To control the agent's movements within the maze, we employ a PID controller. The agent's state, described by a 4-dimensional input including x and y coordinates as well as velocity, informs the policy which then determines the agent's intended location. We explore various maze configurations and controllers for demonstration, and generate demonstrations using two types of planners. The visualization are illustrated in Figure 3. We provide 100 demonstrations for each map.

**Results.**   In Table 1, we present a comparative analysis of BC, Diffusion Policy (DDIM)Chi et al. (2023), and RF-POLICY. Our findings indicate that BC and Diffusion Policy (DDIM) achieve similar performance on success rate, yet they exhibit considerable differences in diversity score. In particular, Diffusion Policy and RF-POLICY display enhanced diversity and are capable of identifying multiple paths to achieve the goal (See Figure4). For RF-POLICY, we visualize the variance predicted by the model in Figure 3. It can be observed that our model learns a reasonable variance that indicates the diversity of actions at a given state. In the Random-Target Maze scenario, both

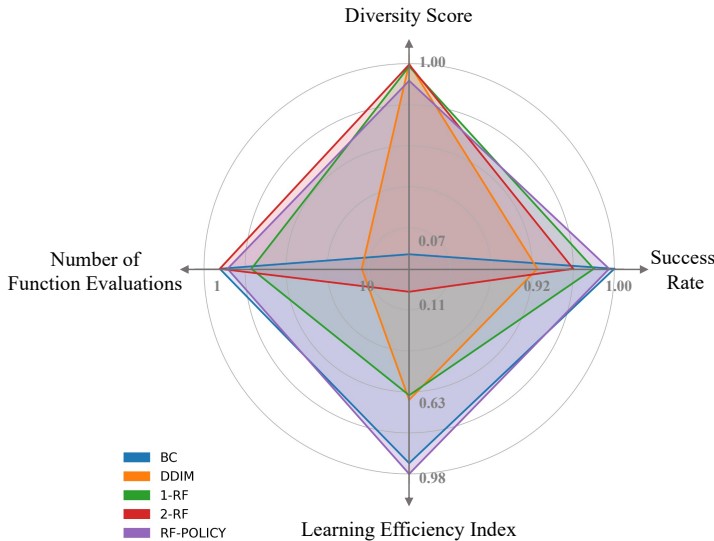

Figure 3: The radar chart compares five methods—Behavioral Cloning (BC), DDIM, 1-Rectified Flow (1-RF), 2-Rectified Flow (2-RF), and RF-POLICY—across four critical metrics: Success Rate, Diversity Score, Learning Efficiency Index (LEI), and Number of Function Evaluations (NFE). RF-POLICY demonstrates balanced and superior performance across all metrics, unlike other methods which exhibit trade-offs between these key performance indicators.

RF-POLICY and DDIM outperform BC, particularly in the more complex Maze 6. This superior performance is likely attributed to the fact that DDIM and RF-POLICY are generative models, which are inherently better suited for learning tasks involving a higher degree of multi-modality.

Table 1: Comparison of Behavioral Cloning (BC), Diffusion Policy (DDIM), and RF-POLICY in Maze tasks. The table showcases the performance metrics including Success Rate (SR), Diversity Score (DS), Number of Function Evaluations (NFE) and Learning Efficiency Index (LEI) for each model across different maze complexities.

| | Maze 1 | | | | Maze 2 | | | | Maze 3 | | | | Maze 4 | | | | Average | | | |
|---|---|---|---|---|---|---|---|---|---|---|---|---|---|---|---|---|---|---|---|---|
| Method | SR | DS | NFE | LEI | SR | DS | NFE | LEI | SR | DS | NFE | LEI | SR | DS | NFE | LEI | SR | DS | NFE | LEI |
| BC | 1.00 | 0.09 | 1.00 | 1.00 | 1.00 | 0.04 | 1.00 | 0.94 | 1.00 | 0.05 | 1.00 | 0.97 | 1.00 | 0.10 | 1.00 | 0.79 | 1.00 | 0.07 | 1.00 | 0.92 |
| DDPM | 0.86 | 1.00 | 20.00 | 0.59 | 0.86 | 1.00 | 20.00 | 0.60 | 0.94 | 1.00 | 20.00 | 0.63 | 0.86 | 1.00 | 20.00 | 0.65 | 0.88 | 1.00 | 20.00 | 0.62 |
| DDIM | 0.84 | 1.00 | 10.00 | 0.55 | 0.92 | 1.00 | 10.00 | 0.66 | 1.00 | 1.00 | 10.00 | 0.64 | 0.94 | 1.00 | 10.00 | 0.68 | 0.92 | 1.00 | 10.00 | 0.63 |
| 1-RF | 0.90 | 1.00 | 10.00 | 0.69 | 1.00 | 1.00 | 10.00 | 0.94 | 1.00 | 1.00 | 10.00 | 0.88 | 1.00 | 1.00 | 10.00 | 0.88 | 0.97 | 1.00 | 10.00 | 0.85 |
| 2-RF | 0.86 | 0.98 | 1.00 | 0.09 | 1.00 | 1.00 | 1.00 | 0.12 | 1.00 | 1.00 | 1.00 | 0.12 | 0.98 | 1.00 | 1.00 | 0.11 | 0.96 | 1.00 | 1.00 | 0.11 |
| RF-POLICY | 1.00 | 0.79 | 1.03 | 0.83 | 1.00 | 0.98 | 1.11 | 0.91 | 1.00 | 0.96 | 1.99 | 0.78 | 0.98 | 0.93 | 1.85 | 0.59 | 0.99 | 0.91 | 1.50 | 0.78 |

**Training and Execution Efficiency.** Diffusion Policy necessitates a training time longer than that required by both BC and RF-POLICY, thereby incurring a higher training cost. Since our model employs adaptive step size in ODE, the majority of the action inference process is accomplished using only a single or two steps in ODE, making the computational cost of RF-POLICY comparable to that of BC. In contrast, the Diffusion Policy demands additional steps for inference, with the time being proportional to the number of inference steps in DDIM. Using a single-step inference is not a feasible option for DDIM, as evidenced by Table 1 and Figure 4.

## 5.3 ROBOT MANIPULATION TASKS

**Environments and Tasks.** We conducted experiment on LIBERO benchmark Liu et al. (2023a) for robot manipulation problem. Our training set encompasses six tasks in the LIBERO Kitchen environment. These tasks involve a variety of object interactions and demand a broad spectrum of motor skills, yielding high-quality, human-teleoperated demonstration data. LIBERO employs Robosuite, a modular robot manipulation simulator. More specifically, we focus on 6 pick-and-place tasks in KITCHEN_SCENE_2 within LIBERO. These tasks, such as placing the bowl on the plate, share the same initial states but are subject to variations in robot and object positions due to

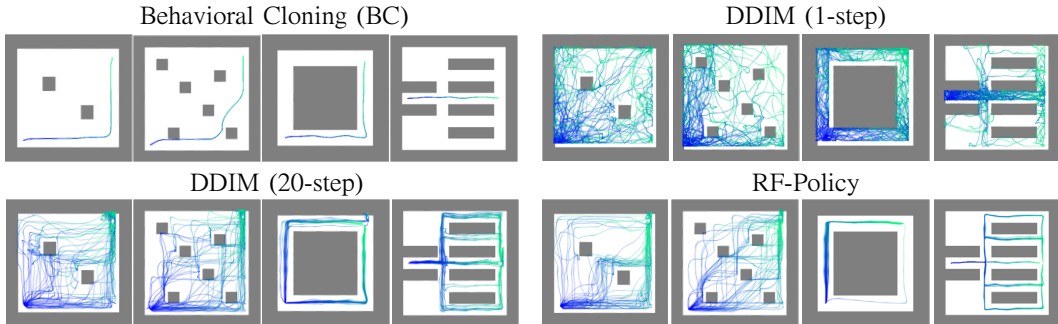

Figure 4: **Generated trajectories.** We visualize the trajectories generated by different policies. The starting point of the agent is kept constant. The results depict that while BC demonstrates a high success rate in navigating the maze, it is limited to learning a single-modal trajectory.

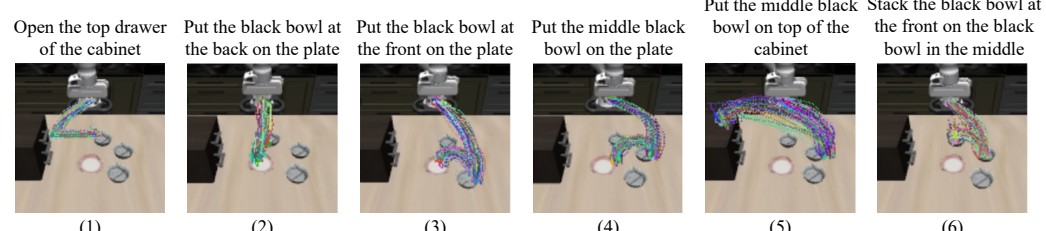

Figure 5: **LIBERO tasks.** We visualize the demonstrated trajectories of the robot's end effector on 6 tasks: (1) Open the top drawer of the cabinet; (2) Put the black bowl at the back on the plate; (3) Put the black bowl at the front on the plate; (4) Put the middle black bowl on the plate; (5) Put the middle black bowl on top of the cabinet; (6) Stack the black bowl at the front on the black bowl in the middle.

noise. For each task, there are 20 demonstration data points. A visualization of the demonstrations for each task is provided in Figure 5.

**Results.** The outcomes on the LIBERO tasks are illustrated in Table 8. Consistent with our observations in the Maze experiment, BC exhibits a lower diversity score compared to DDPM and RF-POLICY. This trend underscores the ability of DDPM and RF-POLICY to generate more varied responses across different tasks, thereby highlighting their adaptability and versatility in diverse scenarios. The consistent performance across different task environments suggests that these models are robust and capable of handling a variety of situations, making them suitable for real-world applications where diversity and adaptability are crucial.

Table 2: Comparison of Behavioral Cloning (BC), Diffusion Policy (DDIM), and RF-POLICY in LIBERO tasks. The table showcases the performance metrics including Success Rate (SR), Diversity Score (DS) and Number of Function Evaluations (NFE) for each model across different task complexities.

| Method | Task 1 | | | Task 2 | | | Task 3 | | | Task 4 | | | Task 5 | | | Task 6 | | | Average | | |
|---|---|---|---|---|---|---|---|---|---|---|---|---|---|---|---|---|---|---|---|---|---|
| | SR | DS | NFE | SR | DS | NFE | SR | DS | NFE | SR | DS | NFE | SR | DS | NFE | SR | DS | NFE | SR | DS | NFE |
| BC | 0.88 | 0.99 | 1.00 | 0.80 | 0.91 | 1.00 | 0.96 | 0.91 | 1.00 | 0.78 | 0.68 | 1.00 | 0.92 | 0.92 | 1.00 | 0.82 | 0.92 | 1.00 | 0.86 | 0.89 | 1.00 |
| DDIM | 0.94 | 0.99 | 10.00 | 0.84 | 0.87 | 10.00 | 0.98 | 0.93 | 10.00 | 0.78 | 0.85 | 10.00 | 0.82 | 0.89 | 10.00 | 0.92 | 0.97 | 10.00 | 0.88 | 0.92 | 10.00 |
| 1-RF | 0.96 | 1.00 | 10.00 | 0.82 | 0.91 | 10.00 | 1.00 | 0.96 | 10.00 | 0.82 | 0.85 | 10.00 | 0.86 | 0.99 | 10.00 | 0.96 | 0.97 | 10.00 | 0.90 | 0.95 | 10.00 |
| 2-RF | 0.90 | 1.00 | 1.00 | 0.82 | 0.89 | 1.00 | 0.98 | 0.96 | 1.00 | 0.82 | 0.86 | 1.00 | 0.82 | 0.93 | 1.00 | 0.96 | 0.98 | 1.00 | 0.88 | 0.94 | 1.00 |
| RF-POLICY | 0.98 | 1.00 | 1.00 | 0.80 | 0.87 | 2.00 | 0.98 | 0.96 | 1.00 | 0.82 | 0.86 | 1.03 | 0.90 | 0.97 | 2.00 | 0.96 | 0.98 | 1.21 | 0.91 | 0.94 | 1.37 |

# 6 CONCLUSION

In conclusion, we present *Rectified Flow-Policy* (RF-POLICY), a novel imitation learning algorithm adept at efficiently generating diverse and adaptive policies, addressing the trade-off between computational efficiency and behavioral diversity inherent in current models. Through extensive experimentation across various settings, RF-POLICY demonstrated superior performance across multiple dimensions including task success rate, behavioral diversity, and training/execution efficiency.

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

## 7 APPENDIX

### 7.1 RESULTS ON ROBOMIMIC

We implement Rectified Flow on RoboMimic Mandlekar et al. (2021) for action generation. The results are shown in Table 3. In our main paper, the implementation follows the Diffusion Policy Chi et al. (2023). For BC model, unlike the approach taken in that work, we use a Behavior Cloning (BC) model with an architecture identical to that of DDPM and DDIM, utilizing L2 regression loss as the loss function. We discovered that when employing the same architecture, BC, Diffusion Policy, and Rectified Flow exhibit comparable performance on RoboMimic. It is noteworthy that, due to resource constraints, we have adjusted the training epochs for the reimplementation of Diffusion Policy to 2000, as opposed to the 3000 epochs used in the original paper; the latter necessitates approximately one month of training.

Table 3: Performance Comparison on RoboMimic Including Reimplemented "Diffusion Policy" Chi et al. (2023)

|  | Lift | | Can | | Square | | Averaged | Push-T |
| --- | --- | --- | --- | --- | --- | --- | --- | --- |
|  | mh | ph | mh | ph | mh | ph | | |
| BC | 1.00 | 1.00 | 0.96 | 1.00 | 0.88 | 0.92 | 0.96 | 0.83 |
| DiffusionPolicy-C (Reimplemented) | 1.00 | 1.00 | 0.96 | 1.00 | 0.94 | 0.92 | 0.97 | 0.96 |
| DiffusionPolicy-C Chi et al. (2023) | 1.00 | 1.00 | 1.00 | 1.00 | 0.98 | 0.98 | 0.99 | 0.95 |
| Rectified Flow | 1.00 | 1.00 | 0.98 | 1.00 | 0.90 | 0.96 | 0.97 | 0.97 |

### 7.2 PLANNER

Similar to Fu et al. (2020), we generate the demonstration data in Maze toy using two types of planners. The planner devises a path in a maze environment by calculating waypoints between the start and target points. It begins by transforming the given continuous-state space into a discretized grid representation. Employing Q-learning, it evaluates the optimal actions and subsequently computes the waypoints by performing a rollout in the grid, introducing random perturbations to the waypoints for diversity. The controller then selects a subset of these waypoints in an ordered manner to form a feasible path. In runtime, it dynamically adjusts the control action based on the proximity to the next waypoint and switches waypoints when close enough, ensuring the trajectory remains adaptive and efficient. The two types of planners operate on the same foundational concept, with the distinction lying in the magnitude of noise introduced: one incorporates smaller noise, while the other employs larger noise in determining the waypoints.

### 7.3 RF-POLICY AS ODE SOLVERS FOR UNCONDITIONAL IMAGE GENERATION

We explored the applicability of RF-POLICY as an adaptive step size ODE solvers to the domain of image generation. We integrated our variance estimation approach with a U-Net-based DDPM++ architecture as in Liu (2022) to generate images on the CIFAR-10 dataset. Performance was evaluated using Frechet Inception Distance (FID).

**Analysis and Results** RF-POLICY facilitates larger step sizes and faster generation speeds when the ODE trajectory is relatively linear. To assess the linearity of image generation trajectories, in Figure 6 (a), we visualized the curvature by computing the cosine similarities between image vectors throughout the generation process and the very first step. The trajectories maintained a near-linear progression with minor deviations occurring between steps 200 and 800. This observation aligns with RF-POLICY's variance predictions shown in Figure 6 (b), which are minimal initially and increase towards the end of the generation process.

In Figure 6 (c), we evaluated RF-POLICY's performance with different threshold parameter $eta$. We found that RF-POLICY's performance is better than the Euler method, but still needs multiple steps to achieve good performance, suggesting that image generation ODEs possess non-linear trajectories. This contrasts with robotics tasks, where deterministic behaviors lead to inherently linear trajectories, making RF-POLICY more effective.

Figure 6: The figure illustrates three key findings: (a) the temporal evolution of cosine similarity across different trajectories, color-coded for clarity; (b) the variance estimations provided by

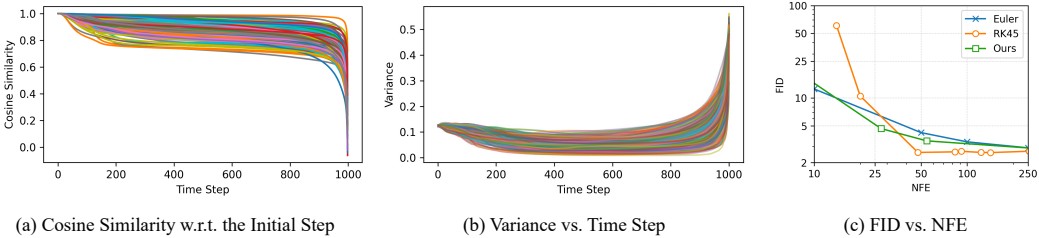

(a) Cosine Similarity w.r.t. the Initial Step    (b) Variance vs. Time Step    (c) FID vs. NFE

Figure 6: The figure illustrates three key findings: (a) the temporal evolution of cosine similarity relative to initial step across different trajectories, color-coded for clarity; (b) the variance estimations provided by RF-POLICY; and (c) the correlation between FID scores and the Number of Function Evaluations (NFE) for the Euler method, Runge–Kutta (RK45) method, and RF-POLICY. For RF-POLICY, we present averaged NFE due to its variable step nature.

VarStep; and (c) the correlation between FID scores and the Number of Function Evaluations (NFE) for the Euler method, Runge–Kutta (RK45) method, and VarStep. For VarStep, we present averaged NFE due to its variable step nature.

## 7.4 COMPARATIVE ANALYSIS OF SEPARATE AND JOINT TRAINING

In this appendix section, we provide a comprehensive comparison between the two training strategies employed in our proposed solution: separate training and joint training. Our primary objective is to investigate whether there is a substantial difference in performance and efficiency between these two training approaches.

**Experiment Setup.** To conduct this comparative analysis, we designed experiments using our proposed framework with both training strategies. Specifically, we consider:

- **Separate Training**: In this approach, we train the variance prediction network and the policy function separately, as described in our main paper.
- **Joint Training**: Here, we train both the variance prediction network and the policy function simultaneously in an end-to-end manner.

The goal is to assess the impact of these training strategies on the overall performance, computational efficiency, and convergence of our model.

**Results and Discussion.** As shown in Table 4, the performance were consistent between the two training approaches, indicating the effectiveness of our two-stage framework in balancing policy accuracy and uncertainty estimation.

Separate training exhibited faster computational speed, making it the preferred choice once the policy function was robustly trained. Joint training required more computational resources and time.

Table 4: Performance comparison of separate training and joint training of RF-POLICYin Maze tasks. The table presents key performance metrics, including Success Rate (SR), Diversity Score (DS), Number of Function Evaluations (NFE), and Learning Efficiency Index (LEI), across various maze complexities. Results are shown for each model in different maze scenarios, as well as the average performance.

| | Maze 1 | | | | Maze 2 | | | | Maze 3 | | | | Maze 4 | | | | Average | | | |
|---|---|---|---|---|---|---|---|---|---|---|---|---|---|---|---|---|---|---|---|---|
| | SR | DS | NFE | LEI | SR | DS | NFE | LEI | SR | DS | NFE | LEI | SR | DS | NFE | LEI | SR | DS | NFE | LEI |
| RF-POLICY (Separate) | 1.00 | 0.98 | 1.11 | 0.91 | 1.00 | 0.79 | 1.03 | 0.83 | 1.00 | 0.96 | 1.99 | 0.78 | 0.98 | 0.93 | 1.85 | 0.59 | 0.99 | 0.91 | 1.50 | 0.78 |
| RF-POLICY (Joint) | 1.00 | 0.97 | 1.09 | 0.67 | 1.00 | 0.93 | 1.08 | 0.76 | 1.00 | 0.91 | 2.08 | 0.64 | 1.00 | 0.91 | 1.70 | 0.60 | 1.00 | 0.93 | 1.49 | 0.67 |

## 7.5 EVALUATING RF-POLICY ON MULTIMODAL 'PICK AND PLACE' TASK

We collected demonstrations for a 'pick and place' task with diverse demonstrations. The 'pick and place' task involves different methods of picking up a bowl, reflecting the diversity of solutions in practical applications. For example, Figure 7 shows two ways of picking up the bowl. This task aligns closely with real-world scenarios, which often require learning from multiple solution strategies.

Table 5: Performance comparison of standard Rectified Flow and RF-POLICY in Maze tasks. We compare with both 1-RF (before reflow) and 2-RF (after reflow). The table presents key performance metrics, including Success Rate (SR), Diversity Score (DS), Number of Function Evaluations (NFE), and Learning Efficiency Index (LEI), across various maze complexities. Results are shown for each model in different maze scenarios, as well as the average performance.

| | Maze 1 | | | | Maze 2 | | | | Maze 3 | | | | Maze 4 | | | | Average | | | |
|---|---|---|---|---|---|---|---|---|---|---|---|---|---|---|---|---|---|---|---|---|
| | SR | DS | NFE | LEI | SR | DS | NFE | LEI | SR | DS | NFE | LEI | SR | DS | NFE | LEI | SR | DS | NFE | LEI |
| 1-RF(step=1) | 1.00 | 0.95 | 1.00 | 0.94 | 1.00 | 0.93 | 1.00 | 0.92 | 1.00 | 0.26 | 1.00 | 0.84 | 0.96 | 0.59 | 1.00 | 0.75 | 0.99 | 0.68 | 1.00 | 0.86 |
| 1-RF(step=5) | 1.00 | 1.00 | 5.00 | 0.93 | 0.92 | 1.00 | 5.00 | 0.75 | 1.00 | 0.99 | 5.00 | 0.83 | 0.92 | 1.00 | 5.00 | 0.84 | 0.96 | 1.00 | 5.00 | 0.84 |
| 1-RF(step=20) | 1.00 | 1.00 | 20.00 | 0.92 | 0.82 | 1.00 | 20.00 | 0.68 | 1.00 | 1.00 | 20.00 | 0.84 | 0.94 | 1.00 | 20.00 | 0.86 | 0.94 | 1.00 | 20.00 | 0.83 |
| 2-RF(step=1) | 1.00 | 1.00 | 1.00 | 0.12 | 0.86 | 0.98 | 1.00 | 0.09 | 1.00 | 1.00 | 1.00 | 0.12 | 0.98 | 1.00 | 1.00 | 0.11 | 0.96 | 1.00 | 1.00 | 0.11 |
| 2-RF(step=5) | 1.00 | 1.00 | 5.00 | 0.12 | 0.86 | 1.00 | 5.00 | 0.10 | 1.00 | 1.00 | 5.00 | 0.12 | 0.98 | 1.00 | 5.00 | 0.11 | 0.96 | 1.00 | 5.00 | 0.11 |
| 2-RF(step=20) | 1.00 | 1.00 | 20.00 | 0.12 | 0.88 | 1.00 | 20.00 | 0.10 | 1.00 | 1.00 | 20.00 | 0.12 | 0.96 | 1.00 | 20.00 | 0.11 | 0.96 | 1.00 | 20.00 | 0.11 |
| RF-POLICY | 1.00 | 0.98 | 1.11 | 0.91 | 1.00 | 0.79 | 1.03 | 0.83 | 1.00 | 0.96 | 1.99 | 0.78 | 0.98 | 0.93 | 1.85 | 0.59 | 0.99 | 0.91 | 1.50 | 0.78 |

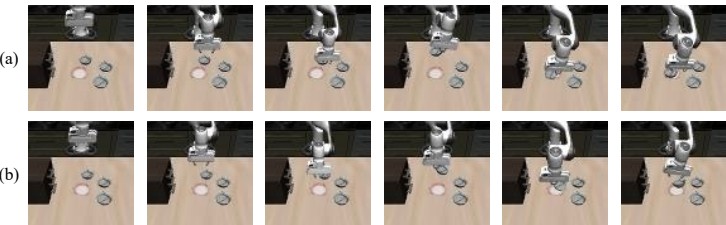

Figure 7: Demonstrating two unique strategies for the 'pick and place' task. In the **top row**, the agent picks up the bowl from its left side. In the **bottom row**, the agent retrieves the bowl from the right side.

We assessed the performance of RF-POLICY against Behavioral Cloning (BC), DDIM and standard Rectified Flow (RF). The results indicated that BC was less adept at capturing the task's multi-modality, resulting in comparatively inferior performance when measured against generative model-based policies. In contrast, RF-POLICY exhibited robust performance, characterized by both a low number of function evaluations (NFE) and high behavioral diversity. This underscores the effectiveness of RF-POLICY in efficiently learning and reproducing diverse strategies integral to complex real-world tasks.

Table 6: Comparison of Behavioral Cloning (BC), Diffusion Policy (DDIM), and RF-POLICY in LIBERO tasks. The table showcases the performance metrics including Success Rate (SR), Diversity Score (DS) and Number of Function Evaluations (NFE) for each model across different task complexities.

| | Task 1 | | | Task 2 | | | Task 3 | | | Task 4 | | | Task 5 | | | Task 6 | | | Average | | |
|---|---|---|---|---|---|---|---|---|---|---|---|---|---|---|---|---|---|---|---|---|---|
| | SR | DS | NFE | SR | DS | NFE | SR | DS | NFE | SR | DS | NFE | SR | DS | NFE | SR | DS | NFE | SR | DS | NFE |
| BC | 0.88 | 0.99 | 1.00 | 0.80 | 0.91 | 1.00 | 0.96 | 0.91 | 1.00 | 0.78 | 0.68 | 1.00 | 0.92 | 0.92 | 1.00 | 0.82 | 0.92 | 1.00 | 0.86 | 0.89 | 1.00 |
| DDIM | 0.94 | 0.99 | 10.00 | 0.84 | 0.87 | 10.00 | 0.98 | 0.93 | 10.00 | 0.78 | 0.85 | 10.00 | 0.82 | 0.89 | 10.00 | 0.92 | 0.97 | 10.00 | 0.88 | 0.92 | 10.00 |
| 1-RF | 0.96 | 1.00 | 10.00 | 0.82 | 0.91 | 10.00 | 1.00 | 0.96 | 10.00 | 0.82 | 0.85 | 10.00 | 0.86 | 0.99 | 10.00 | 0.96 | 0.97 | 10.00 | 0.90 | 0.95 | 10.00 |
| 2-RF | 0.90 | 1.00 | 1.00 | 0.82 | 0.89 | 1.00 | 0.98 | 0.96 | 1.00 | 0.82 | 0.86 | 1.00 | 0.82 | 0.93 | 1.00 | 0.96 | 0.98 | 1.00 | 0.88 | 0.94 | 1.00 |
| RF-POLICY | 0.98 | 1.00 | 1.00 | 0.80 | 0.87 | 2.00 | 0.98 | 0.96 | 1.00 | 0.82 | 0.86 | 1.03 | 0.90 | 0.97 | 2.00 | 0.96 | 0.98 | 1.21 | 0.91 | 0.94 | 1.37 |

### 7.6 COMPARISON OF DIFFERENT METHODS IN LOW NFE CONSTRAINTS.

we present a detailed analysis of RF-POLICY, DDIM, and 1-Rectified Flow (1-RF) across various settings with a limited Number of Function Evaluations (NFE). It can be seen that both 1-RF and DDIM cannot achieve desired performance when generating actions in 1 step.

### 7.7 VISUALIZING PREDICTED VARIANCE IN ROBOTIC TASKS

In this section, we focus on the variance predictions made by RF-POLICY across different states within a robot's state space. Figure 8 presents these predictions, offering a visualization of the diversity in potential actions given a robot's specific state. As illustrated in the figure, for the pick and place task, there is an observable increase in action diversity as the robot's end effector gets closer to the target object. This pattern suggests that while most states exhibit low action variance, implying a uni-modal distribution of potential actions, a subset of states, particularly those in proximity to the object interaction, display a significant increase in variance, reflecting a multi-modal distribution of actions.

Table 7: **Comparative Performance of RF-POLICY, DDIM, and 1-RF Across Different Low-NFE Settings.** This table showcases the Success Rate (SR), Diversity Score (DS), Number of Function Evaluations (NFE), and Learning Efficiency Index (LEI) achieved by each method under a spectrum of NFE constraints.

| | Maze 1 | | | | Maze 2 | | | | Maze 3 | | | | Maze 4 | | | | Average | | | |
|---|---|---|---|---|---|---|---|---|---|---|---|---|---|---|---|---|---|---|---|---|
| | SR | DS | NFE | LEI | SR | DS | NFE | LEI | SR | DS | NFE | LEI | SR | DS | NFE | LEI | SR | DS | NFE | LEI |
| DDIM(step=1) | 0.12 | 1.00 | 1.00 | 0.07 | 0.04 | 1.00 | 1.00 | 0.03 | 0.04 | 1.00 | 1.00 | 0.01 | 0.02 | 1.00 | 1.00 | 0.01 | 0.06 | 1.00 | 1.00 | 0.03 |
| DDIM(step=3) | 0.88 | 1.00 | 3.00 | 0.64 | 0.78 | 1.00 | 3.00 | 0.54 | 0.90 | 1.00 | 3.00 | 0.55 | 0.84 | 1.00 | 3.00 | 0.52 | 0.85 | 1.00 | 3.00 | 0.56 |
| DDIM(step=5) | 0.88 | 1.00 | 5.00 | 0.64 | 0.76 | 1.00 | 5.00 | 0.53 | 0.98 | 1.00 | 5.00 | 0.65 | 0.96 | 1.00 | 5.00 | 0.69 | 0.90 | 1.00 | 5.00 | 0.63 |
| DDIM(step=10) | 0.92 | 1.00 | 10.00 | 0.66 | 0.84 | 1.00 | 10.00 | 0.55 | 1.00 | 1.00 | 10.00 | 0.64 | 0.94 | 1.00 | 10.00 | 0.68 | 0.92 | 1.00 | 10.00 | 0.63 |
| DDIM(step=20) | 0.92 | 1.00 | 20.00 | 0.65 | 0.78 | 1.00 | 20.00 | 0.56 | 0.98 | 1.00 | 20.00 | 0.60 | 0.92 | 1.00 | 20.00 | 0.69 | 0.90 | 1.00 | 20.00 | 0.63 |
| 1-RF(step=1) | 1.00 | 0.95 | 1.00 | 0.94 | 1.00 | 0.93 | 1.00 | 0.92 | 1.00 | 0.26 | 1.00 | 0.84 | 0.96 | 0.59 | 1.00 | 0.75 | 0.99 | 0.68 | 1.00 | 0.86 |
| 1-RF(step=2) | 1.00 | 0.99 | 2.00 | 0.85 | 0.98 | 0.96 | 2.00 | 0.85 | 1.00 | 0.78 | 2.00 | 0.82 | 0.92 | 0.87 | 2.00 | 0.76 | 0.97 | 0.90 | 2.00 | 0.82 |
| 1-RF(step=3) | 1.00 | 0.99 | 3.00 | 0.89 | 0.96 | 0.99 | 3.00 | 0.75 | 1.00 | 0.98 | 3.00 | 0.83 | 0.96 | 0.99 | 3.00 | 0.79 | 0.98 | 0.99 | 3.00 | 0.82 |
| 1-RF(step=5) | 1.00 | 1.00 | 5.00 | 0.93 | 0.92 | 1.00 | 5.00 | 0.75 | 1.00 | 0.99 | 5.00 | 0.83 | 0.92 | 1.00 | 5.00 | 0.84 | 0.96 | 1.00 | 5.00 | 0.84 |
| 1-RF(step=10) | 1.00 | 1.00 | 10.00 | 0.94 | 0.90 | 1.00 | 10.00 | 0.69 | 1.00 | 1.00 | 10.00 | 0.88 | 1.00 | 1.00 | 10.00 | 0.88 | 0.97 | 1.00 | 10.00 | 0.85 |
| 1-RF(step=20) | 1.00 | 1.00 | 20.00 | 0.92 | 0.82 | 1.00 | 20.00 | 0.68 | 1.00 | 1.00 | 20.00 | 0.84 | 0.94 | 1.00 | 20.00 | 0.86 | 0.94 | 1.00 | 20.00 | 0.83 |
| RF-POLICY | 1.00 | 0.98 | 1.11 | 0.91 | 1.00 | 0.79 | 1.03 | 0.83 | 1.00 | 0.96 | 1.99 | 0.78 | 0.98 | 0.93 | 1.85 | 0.59 | 0.99 | 0.91 | 1.50 | 0.78 |

| Open the top drawer of the cabinet | Put the black bowl at the back on the plate | Put the black bowl at the front on the plate | Put the middle black bowl on the plate | Put the middle black bowl on top of the cabinet | Stack the black bowl at the front on the black bowl in the middle |
|---|---|---|---|---|---|

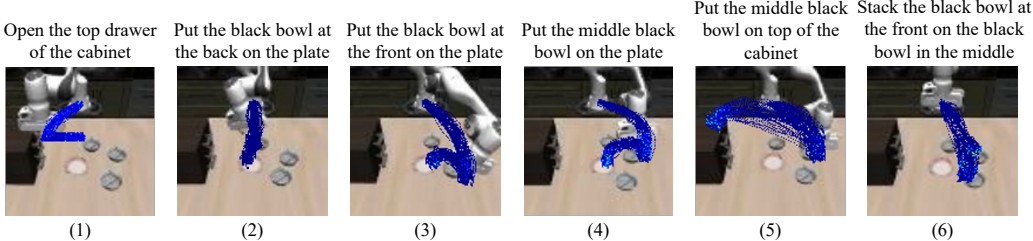

| (1) | (2) | (3) | (4) | (5) | (6) |
|---|---|---|---|---|---|

Figure 8: Visualizations of predicted variance by RF-POLICY in robotics tasks. cooler tones (blue) indicate lower variance states, and warmer tones (red) denote higher variance states.

Table 8: Comparison of Behavioral Cloning (BC), Diffusion Policy (DDIM), and RF-POLICY in Pick and Place tasks. The table showcases the performance metrics including Success Rate (SR), Diversity Score (DS) and Number of Function Evaluations (NFE) for each model.

| | SR | DS | NFE |
|---|---|---|---|
| BC | 0.92 | 0.90 | 1.00 |
| DDIM | 0.96 | 0.97 | 10.00 |
| RF-POLICY | 0.96 | 0.95 | 1.37 |

