# OpenReview forum: "RF-POLICY: Rectified Flows are Adaptive Decision Makers"
_ICLR.cc/2024/Conference — Submitted to ICLR 2024_

### Official Review · Reviewer_KiF7 · 2023-10-24

**Soundness:** 1 poor
**Presentation:** 2 fair
**Contribution:** 3 good
**Rating:** 5
**Confidence:** 4

**Summary:**

This paper addresses the issue of multi-modal policy generation and inference efficiency by proposing a novel offline imitation learning algorithm, RF-POLICY, to achieve a trade-off between policy diversity and model inference efficiency, especially compared to BC and DDIM.
The proposed method is based on Rectified Flow, and its training consists of two stages: one is to optimize the Rectified Flow to ensure the straightness so as to reduce inference time, and the other is to optimize the variance prediction network to determine the uncertainty of state so as to generate diverse policy when state's uncertainty or variance is high. While optimizing the Rectified Flow, it is proved that the model reduces to one-step generators, thus improving training and inference efficiency compared to DDIM.
Experiments on a 2D maze environment and a simulated robot manipulation benchmark suggest that the proposed method can achieve high performance in task success rate, behavioral diversity and inference speed.
The main contribution of this paper is to derive an offline IL algorithm to improve the computational efficiency of diffusion-based policy generators while maintaining their ability of multi-modal policy generation.

**Strengths:**

1. The paper addresses an important problem of the application of diffusion-based policy generators in realistic scenarios, such as robot manipulation.
2. The paper proposes an offline IL method to directly address the issue and provides a proof to explain why the Rectified Flow-based method can improve the training and inference efficiency, and meanwhile, in order to maintain the multi-model policy generation, the paper proposes a training objective by incorporating the state variance estimation into the loss of Rectified Flow optimization.
3. The paper evaluates the proposed method with a new set of evaluation metrics not including the task success rate, but also the behavioral diversity and computational efficiency, to validate the method's ability.
4. Empirical results show that the proposed method can achieve high performance on both computational efficiency and multi-modal policy generation.

**Weaknesses:**

1. Some claims of the paper are not adequately supported with evidence. For example, the experiments evaluate the method on three dimensions only on two benchmark datasets (the third is in the appendix and performance is comparable), so the last claim in the abstract may be doubted. Another example is that Assumption 1 is too strong and there is no evidence nor data distribution visualization to support it, thus the use of the proposed method with real data and non-expert data (e.g., low-return trajectories) is not convincing based on the limited results shown in this paper. However, the focus of this work is to improve the computational efficiency of diffusion-based policy generators, thus more experiments on other datasets as used in Chi (2023) should be conducted, especially the experiments on real-world robot benchmark if possible.
2. The proposed method is compared with only BC and Diffusion Policy, though the baselines are representative in either field.  Performance improvements over other diffusion-based polices are missing, such as Diffusion-QL (Wang et al., 2023), Decision Diffuser (Ajay et al., 2023), Difusion BC (Pearce et al., 2023), etc, so the significance of this paper is doubted.
3. Description of some figures and experimental results is confusing and need further clarity.

References:
[1] Cheng Chi, Siyuan Feng, Yilun Du, Zhenjia Xu, Eric Cousineau, Benjamin Burchfiel, and Shuran Song. Diffusion policy: Visuomotor policy learning via action diffusion. RSS, 2023.
[2] Anurag Ajay, Yilun Du, Abhi Gupta, Joshua B Tenenbaum, Tommi S Jaakkola, and Pulkit Agrawal. Is conditional generative modeling all you need for decision making? ICLR, 2023.
[3] Wang, Zhendong, Jonathan J. Hunt, and Mingyuan Zhou. Diffusion policies as an expressive policy class for offline reinforcement learning. ICLR, 2023.
[4] Tim Pearce, Tabish Rashid, Anssi Kanervisto, David Bignell, Mingfei Sun, Raluca Georgescu, Sergio Valcarcel Macua, Shan Zheng Tan, Ida Momennejad, Katja Hofmann, et al. Imitating human behaviour with diffusion models. ICLR, 2023.

**Questions:**

1. Does the textual description corresponds correctly to the picture in Fig.4?
2. The measurement unit of training and execution time is different as stated in Results in section 5.3, so is it appropriate to exhibit the training and inference efficiency in the same figure as in Fig.5? In addition, the training and execution efficiency of experiments on 2D maze are not quantified clearly.
3. How are the experimental results calculated in Table 1 and Table 2? Are they average scores across several seeds? And some implementation details are missing, such as epochs.
4. In Table 1, results on Maze 5 only exhibit SR and Maze 6 only exhibits DS.

---

> ### Author Response · Authors · 2023-11-23
> **Response**
>
> We sincerely thank the reviewers for their insightful comments and constructive suggestions, which have been instrumental in enhancing the clarity and depth of our work.
>
> **Q1. Assumption 1 and Real Data Application:**
>
> Regarding Assumption 1, we acknowledge the need for empirical support. To this end, we have conducted further analysis to provide data distribution visualizations that substantiate our assumption. These additional results will bolster the credibility of applying RF-POLICY to real and non-expert data.
>
> To substantiate this assumption, we have conducted additional analyses.
> We take advantage of the variance predictions made by RF-POLICY across different states within a robot's state space. Figure 8 in our paper ([link](https://anonymous.4open.science/r/RF-POLICY-184B/libero_variance.png)) presents these predictions, offering a visualization of the diversity in potential actions given a robot's specific state. As illustrated in the figure, for the pick and place task, there is an observable increase in action diversity as the robot's end effector gets closer to the target object. This pattern suggests that while most states exhibit low action variance, implying a uni-modal distribution of potential actions, a subset of states, particularly those in proximity to the object interaction, display a significant increase in variance, reflecting a multi-modal distribution of actions.
>
> **Q2. Comparisons with Chi et. al.:**
>
> We acknowledge the importance of real-world robot experimentation to validate our findings, however, at the current moment the authors do not have real robots to conduct such experiments. As a compensation, we conduct more experiments in simulation (Please see the general response).
>
> Regarding benchmarks from Chi et al.'s work, we have indeed included results from the robomimic benchmark in the appendix of our initial submission (Section 1 in Appendix). Notably, we observed that the success rates for BC, Diffusion Policy, and RF-POLICY were all exceedingly high, consistently above 0.95, indicating a ceiling effect that limits the differentiation between methods. To address this, we shifted our focus to the LIBERO benchmark, which presents more challenging and discriminative tasks.
>
> We plan to undertake a comprehensive set of experiments, including those on RoboMimic, and will incorporate these results in an updated version of our work.
>
> **Q3: Comparison with other offline RL learning methods:**
>
> Our work focuses on offline imitation learning, not offline reinforcement learning, which means the rewards are not present. This is a more practical assumption at least in robotics, since learning complex manipulation/navigation behaviors often requires complex reward design, while providing demonstrations is much simpler. As a result, our method is not directly comparable against methods like Diffusion-QL and Decision Diffuser. Regarding Diffusion BC, it uses DDPM to train a diffusion model for action prediction, this objective is the same as Diffusion Policy we compared with.
>
> In response to the review, we have expanded our comparative analysis to include additional relevant baselines within the scope of imitation learning (See general response). These additional comparisons are better aligned with our approach and further validate the efficacy of RF-POLICY in the intended setting.
>
> We thank the reviewer for pointing out the typos in Figures and tables. We have fixed the typos and revise the descriptions of figures and experimental results for clarity. Please see the updated paper for these change. Considering the time constraints during rebuttal, we will include the average scores across multiple seeds to ensure that the results reflect a robust evaluation of our method.

---

### Official Review · Reviewer_J2ZQ · 2023-10-30

**Soundness:** 3 good
**Presentation:** 4 excellent
**Contribution:** 3 good
**Rating:** 6
**Confidence:** 3

**Summary:**

The paper presents an imitation learning algorithm using rectified flow, a recent advancement in flow-based generative modeling combined with probability flow ordinary differential equations. The main idea is to improve the computational efficiency when action mapping from state is deterministic, in which ODE can be solved trivially. The resulting algorithm generates diverse policies yet avoid unnecessary computation whenever mapping from state to action has sufficiently low variance.

**Strengths:**

1. The proposed algorithm achieves a balance between computational complexity and diversity in generated policies.

**Weaknesses:**

1. The paper lacks empirical or mathematical validation for Assumption 1. The authors posit that most demonstration datasets for robotic tasks exhibit deterministic behavior (or uni-modal states), yet they fail to support this claim with experimental evidence.
2. As delineated by Theorem 1, the RF-Policy loss function (equation 5) optimizes the flow model (ODE model) to generate deterministic (uni-modal) behaviors, evident when the loss function goes to 0 as the variance of action given state reaches 0. This prevents the model to generate multi-modal behaviors. This seems to counter the purpose of using diffusion models.
3. The study omits a comparative analysis with established offline RL baselines such as CQL and BCQ, as well as other diffusion-based methodologies like Diffuser (Janner et al., 2022) and Diffusion-QL (Wang et al., 2022).
4. The paper would benefit from a detailed proof of Theorem 1.

**Questions:**

1. How do linear flow models (linear ODE models), like RF-Policy, accurately encapsulate complex behaviors? Most existing methods have relied on non-linear SDEs, specifically DDPM, for policy estimation, yet this study utilizes a linear model. What rationale is provided for the superiority of this linear approach over its predecessors? (Related to equation 3)
2. Figure 1 is intended to demonstrate that, unlike DDIM (an extension of DDPM), RF-Policy generates straight lines in deterministic areas (x < 0). However, I cannot see distinguishable difference between the red (DDIM) and blue (RF-Policy) lines in the figure. It could be considered a potential weakness of the paper.
3. How does the variance prediction network determine whether a state is uni-modal or multi-modal? It is trained to estimate state variance using an offline dataset, encompassing both epistemic and aleatoric uncertainties. Given that the distinction between uni-modal and multi-modal states pertains to aleatoric uncertainty, how does the model address epistemic uncertainty?

---

> ### Author Response · Authors · 2023-11-23
> **Response**
>
> We sincerely thank the reviewers for their insightful comments and constructive suggestions, which have been instrumental in enhancing the clarity and depth of our work.
>
> **Q1. Validation of Assumption 1:**
>
> To substantiate this assumption, we have conducted additional analyses.
>
> We take advantage of the variance predictions made by RF-POLICY across different states within a robot's state space. Figure 8 ([link](https://anonymous.4open.science/r/RF-POLICY-184B/libero_variance.png)) presents these predictions, offering a visualization of the diversity in potential actions given a robot's specific state. As illustrated in the figure, for the pick and place task, there is an observable increase in action diversity as the robot's end effector gets closer to the target object. This pattern suggests that while most states exhibit low action variance, implying a uni-modal distribution of potential actions, a subset of states, particularly those in proximity to the object interaction, display a significant increase in variance, reflecting a multi-modal distribution of actions.
>
> **Q2: Theorem 1 proof:**
>
> Theorem 1 proves that for deterministic behaviors, where the variance of action given state is zero, the RF-POLICY loss function is optimized such that the flow model will produce straight-line trajectories.
> However, this does not limit RF-POLICY's ability to generate multi-modal behaviors. There’s an expectation over the optimal velocity, $v^*(z, t | x) $, as outlined in the proof, captures the **expected trajectory**.
>
> In the multi-modal case, this expectation accounts for the *multiple* possible directions $y - Z_0$ that can pass through point $z$ at time $t$. Consequently, for a multi-modal distribution, the variance would be greater than zero, allowing for the generation of diverse trajectories.
>
> Our empirical evaluations support the model's multi-modal generative capabilities. In scenarios with inherent uncertainty and multiple valid actions for a given state, RF-POLICY successfully generates a diverse set of plausible actions, which is proof of its ability to model multi-modal behavior effectively.
>
> **Q3: Comparison with other offline imitation learning methods:**
>
> Our work focuses on offline imitation learning, not offline reinforcement learning, which means the rewards are not present. This is a more practical assumption at least in robotics, since learning complex manipulation/navigation behaviors often requires complex reward design, while providing demonstrations is much simpler. As a result, our method is not directly comparable against methods like CQL and BCQ.
>
> In response to the review, we have expanded our comparative analysis to include additional relevant baselines within the scope of imitation learning (See general response). These additional comparisons are better aligned with our approach and further validate the efficacy of RF-POLICY in the intended setting.
>
> **Q4: How does flow model capture complex behaviors?**
>
> RF wants to construct ODEs that follow straight trajectories, which is very different from assuming a linear model. Constructing straight ODE is a non-trivial task, requires quite a lot of non-linearity to achieve: the ODE has to control the move of the particle carefully to ensure the straight trajectory. Mathematically, the velocity field of straight ODEs is characterized by a nonlinear PDE called burger’s equation[1]:
>
> $\partial v/ \partial t + (\partial v/ \partial z) v = 0$, where $v = v(Z_t, t)$ is the velocity field.
>
> This is a challenging and highly non-linear problem to solve. RF seeks for straightness because it is easier for simulate at inference time (more straight => fewer steps).
>
> [1] See Liu, et. al., Flow Straight and Fast. 2023
>
> **Q5. Clarifying Trajectories in Figure 1:**
>
> Figure 1 presents a 1D example with mappings that highlight the behavior of both DDIM and RF-Policy. Although both methods fit the demonstration data well, the red line (DDIM) demonstrates a non-linear trajectory, indicating that multiple inference steps are necessary due to the nature of SDEs. In contrast, the blue line (RF-Policy) presents straight trajectories for deterministic cases (x ≤ 0), validating RF-Policy's ability to efficiently generate actions in a single step through one-step Euler approximation ​

---

> > ### Author Response · Authors · 2023-11-23
> > **Response - Part 2**
> >
> > **Q6:How does the variance prediction network determine whether a state is uni-modal or multi-modal?**
> >
> > We thank the reviewer for bringing up this question. We want to first emphasize that the multi-modality here is a bit different from the term uncertainty. It is possible that the data has no uncertainty at all, but we want to learn a multi-modal behavior (at certain state, the Q(s, a1) = Q(s, a2) >= Q(s, other_a), so both a1 and a2 are optimal actions). Therefore, at this moment, RF-policy, like other offline imitation learning methods (e.g., BC and Diffusion-policy) are not designed to address the epistemic uncertainty. The epistemic uncertainty could be addressed by either adding more data or extending RF-policy to estimate the (s, a) distribution. Both solutions are also applicable to diffusion policy, but we argue that either case only worsen the computation cost. On the contrary, RF-policy is more training/inference efficient, making it a better candidate to scale up to more data or more complex modeling (like world-model learning).

---

### Official Review · Reviewer_Hbo4 · 2023-10-30

**Soundness:** 3 good
**Presentation:** 3 good
**Contribution:** 1 poor
**Rating:** 3
**Confidence:** 5

**Summary:**

Recent papers in offline imitation learning substitute cross-entropy based behavioural cloning with diffusion-based models as a generative model. This paper proposes to use rectified flow instead, a formulation that, still using a mean-squared error objective, forces trajectories of the probability ODE to have no curvature whatsoever, sacrificing some generative accuracy (as it solves a whole family of transport problems) for maximum generation speed with 1 single function evaluation.

**Strengths:**

The paper is well written and straightforward to follow. Besides, it feels clear that combining rectified flow and offline IL should work in practice, based off intuition from the purely supervised RF case.

**Weaknesses:**

However, a big weakness in the paper consists in its novelty and magnitude of its contribution. Specifically, its unique contribution (apart from a log-variance additive regularizer) - compared to any standard diffusion-based offline IL approach - is to use rectified flow for acceleration, since the training time of the procedure is directly proportional to the NFE (number of function evaluations) required to perform inference for the diffusion model. It is indeed the case that rectified flow provides some of the best generative performance at 1 NFE amidst the class of extended diffusion-inspired models; but all that is proven here is that the approximation error in rectified flow is very compatible with the approximation error from offline IL. We feel the argument would be materially stronger if it was demonstrated 1. on a variety of more realistic domains than some of the toy domains (maze) treated here, for instance Atari-100k seeded with expert trajectories, which shouldn't require industrial levels of compute; and 2. most importantly, if an ablation study and exhaustive comparison with the performance of diffusion samplers specifically tailored to the low-NFE regime (UniPC [1], Heun and others [2] for pure samplers, even widening scope Consistency Models [3] or TRACT [4]) was performed. To me figure 5 simply means that DDIM 20 steps was used as baseline. This choice of DDIM feels arbitrary and it's not clear how much relative loss DDIM10, DDIM5 or another sampler would incur, thus minimizing any contribution that claims an NFE speedup. It is also not clear that Rectified Flow is a unique or best solution to this problem, as for say Consistency Models is also a class of diffusion-like models that could claim the same in the IL setting.

[1] Zhao et al, UniPC: A Unified Predictor-Corrector Framework for Fast Sampling of Diffusion Models.
[2] Karras et al, Elucidating the Design Space of Diffusion-Based Generative Models.
[3] Berthelot et al, TRACT: Denoising Diffusion Models with Transitive Closure Time-Distillation.
[4] Song et al, Consistency Models.

**Questions:**

Being cognizant of deadlines and compute constraints, which additional empirical evidence (section 5) could the authors provide in order to bolster their claims ? I would be willing to raise my score if the experiment section were more convincing.

---

> ### Author Response · Authors · 2023-11-23
> **Response**
>
> We sincerely thank the reviewers for their insightful comments and constructive suggestions, which have been instrumental in enhancing the clarity and depth of our work.
>
> **Q1: Addressing More Realistic Domains:**
>
> We appreciate the reviewer's suggestion to test RF-POLICY on more complex domains, such as the Atari-100k benchmark. However, Atari games, including benchmarks like Atari-100k, use a discrete action space. This poses a challenge for diffusion models, and it may require additional methods to approximate the continuous processes that diffusion models use.
>
> In light of this, and to demonstrate the versatility of RF-POLICY in more sophisticated and realistic scenarios, we have extended our experiments to include a 'pick and place' task. This task incorporates a set of demonstrations showcasing different methods of picking up a bowl. We believe this task mirrors real-world applications more closely, as it involves learning multiple viable solutions for a single task – a common requirement in practical settings. This extension of our experiments serves to better validate the model's effectiveness in complex, real-world applications where diverse solution strategies are essential. The following figure shows two ways of picking up the bowl (See this [link](https://anonymous.4open.science/r/RF-POLICY-184B/diverse_libero.png) or Figure 7 in our paper). The table shows the comparison between bc, standard RF, Rectified Flow and RF-POLICY (See this [link](https://anonymous.4open.science/r/RF-POLICY-184B/diverse_libero.png) or Table 8 in our paper).
>
> |            | SR     | DS   | NFE  |
> |------------|--------|------|------|
> | BC         | 0.92   | 0.90 | 1.00 |
> | DDIM       | 0.96   | 0.97 | 10.00|
> | RF-POLICY  | 0.96   | 0.95 | 1.37 |
>
> Table 8: Comparison of Behavioral Cloning (BC), Diffusion Policy (DDIM), and RF-POLICY in Pick and Place tasks. The table includes performance metrics such as Success Rate (SR), Diversity Score (DS), and Number of Function Evaluations (NFE) for each model.
>
> **Q2: Conducting Ablation Studies and Exhaustive Comparisons:**
>
> We pick step=20 in the original paper because that is the number of steps used by Diffusion Policy. We appreciate the reviewer's emphasis on the necessity of a thorough comparison with diffusion samplers optimized for the low-NFE regime. To address this, we have expanded our comparative analysis to include DDIM and standard 1-RF at varying levels of NFEs. The following table presents this expanded evaluation, offering a clearer perspective on the relative performance trade-offs between these methods when operating under constrained NFEs (See this [link](https://anonymous.4open.science/r/RF-POLICY-184B/table7_low_nfes.png) or Table 7 in our paper).

---

> > ### Author Response · Authors · 2023-11-23
> > **Response - Part 2**
> >
> > |               | Maze 1      |       |       |       | Maze 2      |       |       |       | Maze 3      |       |       |       | Maze 4      |       |       |       | Average     |       |       |       |
> > |---------------|-------------|-------|-------|-------|-------------|-------|-------|-------|-------------|-------|-------|-------|-------------|-------|-------|-------|-------------|-------|-------|-------|
> > |               | SR          | DS    | NFE   | LEI   | SR          | DS    | NFE   | LEI   | SR          | DS    | NFE   | LEI   | SR          | DS    | NFE   | LEI   | SR          | DS    | NFE   | LEI   |
> > | DDIM (step=1) | 0.12        | 1.00  | 1.00  | 0.07  | 0.04        | 1.00  | 1.00  | 0.03  | 0.04        | 1.00  | 1.00  | 0.01  | 0.02        | 1.00  | 1.00  | 0.01  | 0.06        | 1.00  | 1.00  | 0.03  |
> > | DDIM (step=3) | 0.88        | 1.00  | 3.00  | 0.64  | 0.78        | 1.00  | 3.00  | 0.54  | 0.90        | 1.00  | 3.00  | 0.55  | 0.84        | 1.00  | 3.00  | 0.52  | 0.85        | 1.00  | 3.00  | 0.56  |
> > | DDIM (step=5) | 0.88        | 1.00  | 5.00  | 0.64  | 0.76        | 1.00  | 5.00  | 0.53  | 0.98        | 1.00  | 5.00  | 0.65  | 0.96        | 1.00  | 5.00  | 0.69  | 0.90        | 1.00  | 5.00  | 0.63  |
> > | DDIM (step=10)| 0.92        | 1.00  | 10.00 | 0.66  | 0.84        | 1.00  | 10.00 | 0.55  | 1.00        | 1.00  | 10.00 | 0.64  | 0.94        | 1.00  | 10.00 | 0.68  | 0.92        | 1.00  | 10.00 | 0.63  |
> > | DDIM (step=20)| 0.92        | 1.00  | 20.00 | 0.65  | 0.78        | 1.00  | 20.00 | 0.56  | 0.98        | 1.00  | 20.00 | 0.60  | 0.92        | 1.00  | 20.00 | 0.69  | 0.90        | 1.00  | 20.00 | 0.63  |
> > | 1-RF (step=1) | 1.00        | 0.95  | 1.00  | 0.94  | 1.00        | 0.93  | 1.00  | 0.92  | 1.00        | 0.26  | 1.00  | 0.84  | 0.96        | 0.59  | 1.00  | 0.75  | 0.99        | 0.68  | 1.00  | 0.86  |
> > | 1-RF (step=2) | 1.00        | 0.99  | 2.00  | 0.85  | 0.98        | 0.96  | 2.00  | 0.85  | 1.00        | 0.78  | 2.00  | 0.82  | 0.92        | 0.87  | 2.00  | 0.76  | 0.97        | 0.90  | 2.00  | 0.82  |
> > | 1-RF (step=3) | 1.00        | 0.99  | 3.00  | 0.89  | 0.96        | 0.99  | 3.00  | 0.75  | 1.00        | 0.98  | 3.00  | 0.83  | 0.96        | 0.99  | 3.00  | 0.79  | 0.98        | 0.99  | 3.00  | 0.82  |
> > | 1-RF (step=5) | 1.00        | 1.00  | 5.00  | 0.93  | 0.92        | 1.00  | 5.00  | 0.75  | 1.00        | 0.99  | 5.00  | 0.83  | 0.92        | 1.00  | 5.00  | 0.84  | 0.96        | 1.00  | 5.00  | 0.84  |
> > | 1-RF (step=10)| 1.00        | 1.00  | 10.00 | 0.94  | 0.90        | 1.00  | 10.00 | 0.69  | 1.00        | 1.00  | 10.00 | 0.88  | 1.00        | 1.00  | 10.00 | 0.88  | 0.97        | 1.00  | 10.00 | 0.85  |
> > | 1-RF (step=20)| 1.00        | 1.00  | 20.00 | 0.92  | 0.82        | 1.00  | 20.00 | 0.68  | 1.00        | 1.00  | 20.00 | 0.84  | 0.94        | 1.00  | 20.00 | 0.86  | 0.94        | 1.00  | 20.00 | 0.83  |
> > | RF-POLICY     | 1.00        | 0.98  | 1.11  | 0.91  | 1.00        | 0.79  | 1.03  | 0.83  | 1.00        | 0.96  | 1.99  | 0.78  | 0.98        | 0.93  | 1.85  | 0.59  | 0.99        | 0.91  | 1.50  | 0.78  |
> >
> > Caption: Comparative Performance of RF-POLICY, DDIM, and 1-RF Across Different Low-NFE Settings. The table showcases the Success Rate (SR), Diversity Score (DS), Number of Function Evaluations (NFE), and Learning Efficiency Index (LEI) achieved by each method under a range of NFE constraints.
> >
> > In addition, We have expanded the application of RF-POLICY to serve as an adaptive step size ODE solver for unconditional image generation, integrating our variance estimation technique with a U-Net-based DDPM++ architecture for image synthesis on the CIFAR-10 dataset. Our performance evaluation used Frechet Inception Distance (FID) as the metric.
> >
> > Our analysis reveals that RF-POLICY allows for larger step sizes and increased generation speeds in image synthesis when the ODE trajectory demonstrates linearity. In addition, it performs better than RK-45 sampler. The result can be seen here ([link](https://anonymous.4open.science/r/RF-POLICY-184B/cifar10.png) or Figure 6 in our paper).

---

> > > ### Author Response · Authors · 2023-11-23
> > > **Reponse - Part 3**
> > >
> > > **Q3. Exploring Alternatives to DDIM:**
> > >
> > > It is important to note that RF-POLICY is not merely an application of existing rectified flow techniques but is a novel approach, as we explain in the general response. While we acknowledge the potential of consistency models and similar diffusion-like techniques in imitation learning settings, our method stands out due to its efficient training and inference capabilities, as well as its adeptness at learning a spectrum of behaviors—features that are critical for practical, real-world applications.
> > >
> > > Furthermore, consistency models require distillation approaches to achieve low-NFE performance. RF-POLICY, on the other hand, achieves this efficiency intrinsically, without the need for such additional distillation or reflow steps. Future work may explore the possibility of using consistency models, but our current focus is to highlight the unique contributions of RF-POLICY and its practical implications.

---

### Official Review · Reviewer_zCJH · 2023-10-31

**Soundness:** 3 good
**Presentation:** 3 good
**Contribution:** 3 good
**Rating:** 5
**Confidence:** 4

**Summary:**

The paper introduces RF-POLICY, an imitation learning algorithm that leverages Rectified Flow, a recent advancement in flow-based generative modeling. Traditional methods like Behavioral Cloning (BC) struggle with diverse behaviors, and while diffusion-based imitation learning addresses this, it does so at the cost of slower inference. RF-POLICY uniquely employs probability flow ordinary differential equations (ODEs) for policy generation, ensuring rapid inference without compromising on behavioral diversity. The core advantage of RF-POLICY is its adaptability: for uni-modal states, it behaves like a one-step generator, and for multi-modal states, it uses a multi-step approach. Empirical evaluations demonstrate that RF-POLICY offers superior performance across multiple metrics like success rate, behavioral diversity, and inference speed.

**Strengths:**

1. RF-POLICY introduces a novel application of Rectified Flow in imitation learning, highlighting an adaptive mechanism to control generation efficiency based on demonstration variance.

2. RF-POLICY efficiently addresses the trade-off between inference speed and behavioral diversity, which has been a challenge in traditional methods like BC and diffusion-based imitation learning.

3. The paper not only introduces new evaluation metrics for imitation learning but also presents a detailed empirical analysis, demonstrating the algorithm's superior performance across various robotic problems.

4. RF-POLICY is highlighted for its straightforward implementation and rapid training, providing practical advantages in real-world applications.

**Weaknesses:**

1. There is a theoretical gap between the objective at eq.~(8) and the implementation at Alg.1. The implementation uses a rectified flow to train the policy function, and uses another neural network to train the variance prediction network. In execution, the variance prediction network is used to determine the update iteration.
2. Considering that the variance prediction network and the policy are trained separately, the performance gain especially in training is only a contribution of the rectified flow instead of the proposed solution as a whole.

**Questions:**

The paper is clearly written with good visualizations. However, the gap between the objective at eq.(8) and the implementation is not explained.
1. I was wondering if there are any reasons supporting the implementation.
2. I was wondering if using rectified flow to replace the DDPM in the DDIM models will lead to similar performances in both tasks.

---

> ### Author Response · Authors · 2023-11-23
> **Response**
>
> We sincerely thank the reviewers for their insightful comments and constructive suggestions, which have been instrumental in enhancing the clarity and depth of our work.
>
> **Q1. Gap between the objective at Eq.(8) and the implementation at Alg.1**
>
> The two-stage training framework is indeed deliberate and advantageous.
>
> Firstly, training the variance prediction network in fact takes a very short time compared to the training of the RF network. Training the rectified flow model requires a duration that is 10 times longer than that needed to train the variance prediction model.
>
> Secondly, we empirically found that training the two components separately leads to slightly better performance than training them jointly (See the Table here ([link](https://anonymous.4open.science/r/RF-POLICY-184B/table4_separate_joint_train.png)), or Table 4 in the paper).
>
> |                    | Maze 1      |            |       |      | Maze 2      |            |       |      | Maze 3      |            |       |      | Maze 4      |            |       |      | Average     |            |       |      |
> |--------------------|-------------|------------|-------|------|-------------|------------|-------|------|-------------|------------|-------|------|-------------|------------|-------|------|-------------|------------|-------|------|
> |                    | SR          | DS         | NFE   | LEI  | SR          | DS         | NFE   | LEI  | SR          | DS         | NFE   | LEI  | SR          | DS         | NFE   | LEI  | SR          | DS         | NFE   | LEI  |
> | RF-POLICY (Separate) | 1.00        | 0.98       | 1.11  | 0.91 | 1.00        | 0.79       | 1.03  | 0.83 | 1.00        | 0.96       | 1.99  | 0.78 | 0.98        | 0.93       | 1.85  | 0.59 | 0.99        | 0.91       | 1.50  | 0.78 |
> | RF-POLICY (Joint)    | 1.00        | 0.97       | 1.09  | 0.67 | 1.00        | 0.93       | 1.08  | 0.76 | 1.00        | 0.91       | 2.08  | 0.64 | 1.00        | 0.91       | 1.70  | 0.60 | 1.00        | 0.93       | 1.49  | 0.67 |
>
> Table 4: Performance comparison of separate training and joint training of RF-POLICY in Maze tasks. The table presents key performance metrics, including Success Rate (SR), Diversity Score (DS), Number of Function Evaluations (NFE), and Learning Efficiency Index (LEI), across various maze complexities. Results are shown for each model in different maze scenarios, as well as the average performance.
>
> Therefore, though in theory training the two networks jointly or sequentially should be the same, in practice, training two “tasks” simultaneously can lead to optimization challenges (See PCGrad for the conflicting gradient issue when more objectives are introduced). Moreover, as we found that once the RF net is trained, training the variance network adds negligble extra overhead, so we choose this way of training. We have modified the paper to illustrate the reason in detail.
>
> **Q2:  Performance gain in training:**
>
> First, regarding the performance in terms of the success rate and diversity, the benefit indeed comes from the RF network. It is in the original theory of RF that, if we ignore the simulation speed (how many steps to simulate the ODE), RF without any reflow performs the best. Applying RF on robotics problems is similar to applying DDIM/DDPM in the diffusion-policy paper. Moreover, we believe that RF is simpler to implement and easier to undertand, and is more computationally efficient.
>
> However, we would like to argue that regarding the performance in terms of inference speed, the contribution mainly comes from the **variance network**. This variance network helps reduce the ODE simulation steps whenever possible. For instance, if a state always leads to a deterministic action, RF-policy will be as efficient as BC. And the number of simulation steps completely depends on the “variance” (or diversity) of the optimal actions given the state. This part alone is a core contribution.

---

> > ### Author Response · Authors · 2023-11-23
> > **Response - Part 2**
> >
> > **Q3. Replacing DDPM with rectified flow in DDIM models:**
> >
> > In response to your query, we have implemented the standard rectified flow method as an alternative to DDIM in our experiments. Our comparison includes both the 1-Rectified Flow (1-RF, before reflow) and the 2-Rectified Flow (2-RF, after reflow), using a standard Euler sampler for action generation. (See the Table here ([link](https://anonymous.4open.science/r/RF-POLICY-184B/table5_compare_with_rf.png)), or Table 5 in the paper).
> >
> > Our findings indicate that while 1-RF shows limitations in capturing diverse behaviors in a single step, 2-RF demonstrates an ability to produce varied actions with just one step. However, it's important to note that 2-RF involves a reflow process which is approximately 7 times more resource-intensive in terms of training compared to 1-RF, Behavioral Cloning (BC), or our RF-POLICY method.
> >
> > |                  | Maze 1 |      |      |      | Maze 2 |      |      |      | Maze 3 |      |      |      | Maze 4 |      |      |      | Average |      |      |      |
> > |------------------|--------|------|------|------|--------|------|------|------|--------|------|------|------|--------|------|------|------|---------|------|------|------|
> > |                  | SR     | DS   | NFE  | LEI  | SR     | DS   | NFE  | LEI  | SR     | DS   | NFE  | LEI  | SR     | DS   | NFE  | LEI  | SR      | DS   | NFE  | LEI  |
> > | 1-RF (step=1)    | 1.00   | 0.95 | 1.00 | 0.94 | 1.00   | 0.93 | 1.00 | 0.92 | 1.00   | 0.26 | 1.00 | 0.84 | 0.96   | 0.59 | 1.00 | 0.75 | 0.99    | 0.68 | 1.00 | 0.86 |
> > | 1-RF (step=5)    | 1.00   | 1.00 | 5.00 | 0.93 | 0.92   | 1.00 | 5.00 | 0.75 | 1.00   | 0.99 | 5.00 | 0.83 | 0.92   | 1.00 | 5.00 | 0.84 | 0.96    | 1.00 | 5.00 | 0.84 |
> > | 1-RF (step=20)   | 1.00   | 1.00 | 20.00| 0.92 | 0.82   | 1.00 | 20.00| 0.68 | 1.00   | 1.00 | 20.00| 0.84 | 0.94   | 1.00 | 20.00| 0.86 | 0.94    | 1.00 | 20.00| 0.83 |
> > | 2-RF (step=1)    | 1.00   | 1.00 | 1.00 | 0.12 | 0.86   | 0.98 | 1.00 | 0.09 | 1.00   | 1.00 | 1.00 | 0.12 | 0.98   | 1.00 | 1.00 | 0.11 | 0.96    | 1.00 | 1.00 | 0.11 |
> > | 2-RF (step=5)    | 1.00   | 1.00 | 5.00 | 0.12 | 0.86   | 1.00 | 5.00 | 0.10 | 1.00   | 1.00 | 5.00 | 0.12 | 0.98   | 1.00 | 5.00 | 0.11 | 0.96    | 1.00 | 5.00 | 0.11 |
> > | 2-RF (step=20)   | 1.00   | 1.00 | 20.00| 0.12 | 0.88   | 1.00 | 20.00| 0.10 | 1.00   | 1.00 | 20.00| 0.12 | 0.96   | 1.00 | 20.00| 0.11 | 0.96    | 1.00 | 20.00| 0.11 |
> > | RF-POLICY        | 1.00   | 0.98 | 1.11 | 0.91 | 1.00   | 0.79 | 1.03 | 0.83 | 1.00   | 0.96 | 1.99 | 0.78 | 0.98   | 0.93 | 1.85 | 0.59 | 0.99    | 0.91 | 1.50 | 0.78 |

---

### Author Response · Authors · 2023-11-23
**General Response to Reviewers**

We are grateful for the insightful feedback from the reviewers. This response provides a general overview of the modifications made to our methodology and a common response to all reviewers. We address reviewer-specific questions in individual responses.

**Improved Utilization of the Variance Network (an update on RF-Policy methodology)**:
Per the rebuttal time, the authors have developed a slightly updated method to leverage the variance network ($\sigma$) based on a more solid and deeper theoretical understanding. In addition to the initial manuscript—where $\sigma$ was used to decide whether a state’s trajectory should be 'straight' (modeled with a 1-step ODE simulation) or 'curved' (requiring a multi-step ODE simulation)—our revised methodology extends its application to any given state s and at any time t during the ODE simulation. The detailed algorithm is highlighted in Algorithm box 1 & 2 ([link](https://anonymous.4open.science/r/RF-POLICY-184B/algo1_and_algo2.png)). The main idea is the following:

The predicted variance $\sigma$ could suggest the step size to take at simulation time $t$, such that we can make dynamic adjustments to the ODE steps, thereby speeding the ODE simulation process. The updated method circumvents the necessity for rigid thresholding and establishes a theoretical significance for  $\sigma$. It effectively quantifies the error within ODE simulations, offering a theoretical bound that enhances the accuracy of our simulated outcomes. (See Theorem 2 in the updated paper). Note that this is an extension of the original proposed method that enables further control of the ODE simulation, not a completely different method.

**The Novelty of RF-Policy**:
RF-policy, though inspired by rectified-flow (RF), is quite different from the original RF in three ways:

1. RF-policy does not require the reflow process (the distillation process to straightening the ODE). In standard RF, they first do random matching between noise and target to fit an ODE, then perform reflow to straighten the learned ODE. The reflow process is often much more costly than the initial learning of the ODE. Note that this is the same for the consistency models (Song et al. Consistency Models, 2023). The reflow process is necessary in standard RF because otherwise it takes many Euler steps to simulate the learned ODE during inference. By contrast, RF-POLICY only leverages the initially learned ODE, whose training and inference costs are similar to that of behavioral cloning. As we scale to larger datasets, we believe RF-POLICY can be significantly more useful than the existing diffusion-based approach.

2. RF-POLICY is a conditional generator (e.g., generating actions conditioned on states), and the core observation is that **different states require different levels of action modalities**. While some reviewers raise concern about this observation, note that in practice, even BC (the unimodal policy) can perform reasonably well. This implicitly means that most states tend to produce unimodal output. More concretely, we add an additional visualization of the variance predicted by the variance network in Figure 8 ([link](https://anonymous.4open.science/r/RF-POLICY-184B/libero_variance.png)). From the plot, one can see that is indeed the case that the majority of states lead to less multimodal actions.
As a result, we observe that the performance benefit of diffusion-based methods comes from only a subset of states where the action choices are much more diverse. But note that existing diffusion-based methods, like diffusion policy, are agnostic to whether a state leads to more multimodal or unimodal actions. So even if the optimal demonstration is indeed deterministic, diffusion policy will be equally costly to apply as if the demonstration is highly multimodal. We would like to emphasize that, with the updated method of using the variance network, **the RF-POLICY is adaptive to the extent of multimodality of actions**. If a state is more multimodal, RF-policy simulates a bit slower for a more diverse action set, and if a state is more unimodal, RF-policy simulates the action faster.

3. The variance network serves as the adaptive ODE-solver that determines the stepsize for simulating the ODE. This is **a completely new method based on solid theoretical results**. We show that with this method, the number of simulation steps is completely determined by the “level of multimodality” of the actions. In addition, **the generation error is strictly bounded**.

In addition to its difference from the original RF, RF-POLICY is specially designed just for robot learning (due to the nature of the problem). Here, we show a lidar plot ([link](https://anonymous.4open.science/r/RF-POLICY-184B/maze_radar.png)) to demonstrate that RF-POLICY, so far, achieves the best trade-off among training efficiency, inference efficiency, evaluation performance, and behavior diversity.

---

> ### Author Response · Authors · 2023-11-23
> **General Response to Reviewers - Part 2**
>
> **Theory**:
> 1. In the original manuscript, we have established the result that RF-POLICY yields a straight-line ODE process if $P(a | s)$ is deterministic.
> 2. In the updated method, we present a further result showcasing that the number of simulation steps is determined by how “multimodal” or “diverse” $P(a | s)$ is. And the resulting simulation error is strictly bounded. Note that 2 is a compliment result of 1 and not contradicting 1.
>
> **Comparison to Offline RL diffusion-based methods**:
> Reviewer J2ZQ and KiF7 were curious about how RF-POLICY compares against offline RL methods like the Diffuser, Decision Diffuser, CQL, Decision Diffuser, etc. Here we note that in principle, we can definitely extend RF-policy’s idea to the offline RL setting, but RF-POLICY, just like Diffusion Policy, is a discriminative model that only predicts $P(a | s)$. There is no assumption about if offline rewards are provided, nor do we try to predict the joint (s, a) distribution. Hence, RF-POLICY in principle still remains to be an offline imitation learning method (offline as opposed to online method like GAIL/ AIRL that needs interaction with the environment).
>
> **Empirical Validation:**
> In response to reviewers' feedback, we have conducted additional experiments, which encompass the following aspects:
> 1. *Comparison with Standard RF Method*: We have performed a thorough comparison with the standard Random Forest (RF) method, considering both 1-RF (before reflow) and 2-RF (after reflow) models.
> 2. *Ablation Analysis with Low-NFE*: We have extended our ablation study to include results using low Neural Function Evaluation (NFE) for both DDIM and the standard RF method.
> 3. *Efficiency Metrics for Training and Inference*: We have introduced metrics to assess the efficiency of our models during both training and inference stages.
> 4. *Empirical Validation of Assumption 1*: We have provided empirical validation for Assumption 1 by visualizing the variance in actions observed in robot demonstrations.
> 5. *Multi-Modal Behavior in Robot Manipulation*: We have gathered demonstrations for a pick-and-place robot manipulation task, where the robot employs various poses to pick up objects. We subsequently compared different methods in this multi-modal context.
> 6. *Comparison with Other Sampling Methods:* In the domain of image generation, we have conducted a comparison between RF-POLICY and alternative sampling methods.
>
> We have reflected the extended experiments in the updated version of the submission. Please see the appendix for extended experiments.

---

### Meta-Review · Area_Chair_qTvr · 2023-12-11

**Metareview:**

This paper suggests imitation learning algorithm based on rectified flows. The goal is to achieve more efficient inference building upon the as-straight-as-possible sampling trajectories property of rectified flows. A main observation of the proposed approach is that since typical imitation learning is uni-model (i.e., deterministic - give $x$ predict a unique $y$) then rectified flows provides in this case (ideally) a one step generator. To facilitate more efficient sampling based on this observation the paper learns a variance prediction model and use it to set the step size in the sampling integration procedure. While the paper shows some initial promising results the reviewers felt it has somewhat limited novelty and magnitude of the contribution in applying rectified flows in the context of RL, and its main assumption might be not sufficiently substantiated. The authors are encouraged to include more experiments (as indicated in reviews) as well as including other baselines for solvers and other (NFE efficient) diffusion-like and offline RL methods.

**Justification For Why Not Higher Score:**

Please see the Meta Review for the relevant justification.

**Justification For Why Not Lower Score:**

N/A

---

### Decision · Program_Chairs · 2024-01-16

Reject